# Chemical genetic identification of CDKL5 substrates reveals its role in neuronal microtubule dynamics

Lucas L Baltussen[1,†] (ID), Priscilla D Negraes[2,†] (ID), Margaux Silvestre[1] (ID), Suzanne Claxton[1], Max Moeskops[1], Evangelos Christodoulou[3] (ID), Helen R Flynn[4] (ID), Ambrosius P Snijders[4], Alysson R Muotri[2,5,*] (ID) & Sila K Ultanir[1,**] (ID)

## Abstract

Loss-of-function mutations in CDKL5 kinase cause severe neurodevelopmental delay and early-onset seizures. Identification of CDKL5 substrates is key to understanding its function. Using chemical genetics, we found that CDKL5 phosphorylates three microtubule-associated proteins: MAP1S, EB2 and ARHGEF2, and determined the phosphorylation sites. Substrate phosphorylations are greatly reduced in CDKL5 knockout mice, verifying these as physiological substrates. In CDKL5 knockout mouse neurons, dendritic microtubules have longer EB3-labelled plus-end growth duration and these altered dynamics are rescued by reduction of MAP1S levels through shRNA expression, indicating that CDKL5 regulates microtubule dynamics via phosphorylation of MAP1S. We show that phosphorylation by CDKL5 is required for MAP1S dissociation from microtubules. Additionally, anterograde cargo trafficking is compromised in CDKL5 knockout mouse dendrites. Finally, EB2 phosphorylation is reduced in patient-derived human neurons. Our results reveal a novel activity-dependent molecular pathway in dendritic microtubule regulation and suggest a pathological mechanism which may contribute to CDKL5 deficiency disorder.

**Keywords** CDKL5; chemical genetics; EB2; MAP1S; microtubule dynamics
**Subject Categories** Cell Adhesion, Polarity & Cytoskeleton; Genetics, Gene Therapy & Genetic Disease; Post-translational Modifications, Proteolysis & Proteomics
**The EMBO Journal (2018) 37: e99763**

See also: **IM Muñoz et al** (December 2018) and **PA Eyers** (December 2018)

## Introduction

Cyclin-dependent kinase-like 5 (CDKL5) is an X-linked gene encoding a 115 kDa serine/threonine kinase, a member of the CMGC (CDK, MAPK, GSK3 and CLK containing) kinase family (Manning *et al*, 2002). Mutations in a single copy of the *CDKL5* gene cause a rare neurodevelopmental disorder predominantly observed in girls (Kalscheuer *et al*, 2003; Tao *et al*, 2004; Weaving *et al*, 2004; Scala *et al*, 2005). Although initially categorized as an atypical Rett syndrome, CDKL5 deficiency is now considered a distinct disorder, named CDKL5 deficiency disorder (CDD). CDD is characterized by early-onset seizures and widespread neurodevelopmental delay (Bahi-Buisson & Bienvenu, 2012; Fehr *et al*, 2013). Pathological deletions and nonsense mutations are found throughout the *CDKL5* gene, whereas disease-causing missense variants are highly enriched in the N-terminal kinase domain of CDKL5, highlighting the importance of its kinase activity (Fehr *et al*, 2016).

In mice, CDKL5 is expressed throughout the brain starting at late embryonic stages and through adulthood, indicating a role in postnatal development and function in the nervous system (Chen *et al*, 2010; Hector *et al*, 2016, 2017). CDKL5 mRNA is highly enriched in foetal and adult human brains (Hector *et al*, 2016). CDKL5's functions in excitatory and inhibitory synaptic transmission as well as dendrite, axon and spine development were studied using shRNA-mediated knockdown or CDKL5 knockout mice (Baltussen *et al*, 2017; Zhou *et al*, 2017). In addition, patient-derived induced pluripotent stem cells (hiPSCs) were used to obtain postmitotic neurons from patients with CDKL5 mutations (Amenduni *et al*, 2011; Ricciardi *et al*, 2012). However, no clear consensus has emerged on the cellular and molecular functions of CDKL5. Specifically, its direct physiological substrates are unknown.

1 Kinases and Brain Development Laboratory, The Francis Crick Institute, London, UK
2 Department of Pediatrics, School of Medicine, University of California San Diego, La Jolla, CA, USA
3 Structural Biology Science Technology Platform, The Francis Crick Institute, London, UK
4 Proteomics Science Technology Platform, The Francis Crick Institute, London, UK
5 Department of Pediatrics/Cellular & Molecular Medicine, Center for Academic Research and Training in Anthropogeny (CARTA), Kavli Institute for Brain and Mind, School of Medicine, Rady Children's Hospital San Diego, University of California San Diego, La Jolla, CA, USA
*Corresponding author. Tel: +1 858 534 9320; E-mail: muotri@ucsd.edu
**Corresponding author. Tel: +44 20 3796 1613; E-mail: sila.ultanir@crick.ac.uk
†These authors contributed equally to this work

The microtubule cytoskeleton is required for fundamental cellular processes such as long-range intracellular transport and for dynamic cellular changes such as neural migration or neurite formation (Conde & Caceres, 2009; Kapitein & Hoogenraad, 2015). Microtubules in dendrites are oriented in mixed polarity (Baas *et al*, 1988). Microtubule plus-end growth visualized by fluorescently tagged end-binding proteins in living neurons helped characterize microtubule dynamics in dendrites (Stepanova *et al*, 2003; van de Willige *et al*, 2016; Yau *et al*, 2016). In primary neurons, dendritic neuronal activity and synaptic plasticity manipulations alter microtubule dynamics (Hu *et al*, 2008; Jaworski *et al*, 2009; Kapitein *et al*, 2011; Merriam *et al*, 2011; Ghiretti *et al*, 2016). Microtubule architecture and dynamics influence cargo trafficking on dendrites and axons (Yogev *et al*, 2016; Tas *et al*, 2017). Microtubules play central roles during neuronal development and function, and however, signalling pathways that regulate microtubule architecture, dynamics and microtubule-dependent trafficking remain to be elucidated.

In this study, we identify CDKL5's direct substrates in mouse brain using a chemical genetic method (Hertz *et al*, 2010; Ultanir *et al*, 2012, 2014). We find that CDKL5 phosphorylates three microtubule-binding proteins, Rho guanine nucleotide exchange factor 2 (ARHGEF2), microtubule-associated protein 1S (MAP1S) and microtubule-associated protein RP/EB family member 2 (MAPRE2/EB2), implicating CDKL5 in microtubule-dependent processes. We show that EB2 Ser222 and MAP1S Ser812 phosphorylations are dramatically reduced in the CDKL5 knockout mouse brain, making them *bona fide* phosphorylation sites for CDKL5. EB2 phosphorylation is detected predominantly in dendrites of excitatory neurons. By imaging dendritic microtubule dynamics using EB3-tdTomato, we show that CDKL5 KO neurons have enhanced duration of plus-end growth, an effect that is dependent on MAP1S's direct binding to microtubules. We find specific defects in anterograde cargo trafficking of TrkB puncta. Interestingly, novel substrate phosphorylation is also highly reduced in patient iPSC-derived neurons, indicating that these substrates are conserved and are affected in human patients. Finally, we provide evidence that CDKL5-dependent phosphorylations are NMDAR activity dependent, suggesting a role in activity-dependent circuit formation. Our findings describe a novel regulatory mechanism on microtubules that is compromised in CDKL5 deficiency disorder.

# Results

## Chemical genetic identification of CDKL5 substrates MAP1S, EB2 and ARHGEF2

We used a chemical genetic approach to determine CDKL5's substrates in mouse brain lysates (Fig 1A; Blethrow *et al*, 2004; Hertz *et al*, 2010). This method utilizes analog-specific (AS) kinases, which can accept bulky ATP analogs owing to their enlarged ATP binding pocket. The gamma phosphate in bulky ATP analogs is replaced with a thiophosphate allowing the AS-kinase to thiophosphorylate its substrates. Bio-orthogonal bulky ATPγS analogs cannot be effectively used by wild-type kinases present in biological samples, allowing thiophosphorylation and labelling to be specific for the AS-kinase. First, to obtain an enlarged ATP binding pocket, we mutated the gatekeeper residue Phenylalanine 89 to a smaller amino acid Alanine (F89A) in CDKL5 kinase domain (amino acids 1–352; Appendix Fig S1A; Shah *et al*, 1997). Next, we tested the mutation's effect on CDKL5's autophosphorylation activity using a non-radioactive kinase assay utilizing ATPγS and its analog benzyl-ATPγS. Thiophosphate is reacted with p-nitro-benzyl-mesylate to obtain an epitope that is detected by a thiophosphate ester antibody (Allen *et al*, 2007). We found that F89A mutation enabled CDKL5 to utilize benzyl-ATPγS, albeit at low levels (Fig 1B and C). We engineered a second-site suppressor mutation C152A, which had been used to rescue NDR1 kinase (Ultanir *et al*, 2012; Appendix Fig S1B), and found that CDKL5 F89A/C152A activity with benzyl-ATPγS was comparable to wild-type kinase activity utilizing ATPγS (Fig 1B and C). CDKL5[1–352] F89A/C152A double mutant (from here onwards referred to as AS-CDKL5) could also thiophosphorylate Amphiphysin-1 (AMPH1), confirming its activity towards a known *in vitro* substrate (Sekiguchi *et al*, 2013; Appendix Fig S1C). We purified AS-CDKL5 either from HEK293 cells or from insect cells.

In order to label CDKL5 substrates, we reacted purified AS-CDKL5 and benzyl-ATPγS with 1–2 mg of postnatal day 12 (P12) mouse brain lysate. Thiophosphorylated substrate proteins were digested to peptide fragments, and thiophosphopeptides were purified and identified using mass spectrometry (Fig 1A). We used kinase-dead (KD) CDKL5 (K42A) as negative control and conducted three independent experiments totalling eight AS replicates. We repeatedly detected ARHGEF2 Ser122, EB2 Ser222 and MAP1S

**Figure 1. Identification of CDKL5 substrates using chemical genetics.**

A   Schematic of chemical genetic kinase substrate identification method. AS-CDKL5 utilizes benzyl-ATPγS to thiophosphorylate its substrates. Following trypsin digest, thiol-containing peptides are captured on iodoacetyl agarose beads. Thiophosphate ester linked peptides are released by oxone induced hydrolysis while cysteine-containing peptides remain attached. Eluted phosphopeptides are analysed with liquid chromatography-tandem mass spectrometry (LC-MS/MS).

B, C   CDKL5[1–352] autophosphorylation is shown. WT CDKL5 uses ATPγS, KD is inactive, F89A uses benzyl-ATPγS with reduced efficiency, F89A/C152A mutant has rescued activity. Quantification of Thio-P signal is normalized for CDKL5 levels (HA), and KD background is subtracted. Fisher's LSD: *n* = 4 replicates. bn, benzyl.

D   A list of novel CDKL5 substrate targets and phosphorylation sites identified. All phosphorylation sites are conserved in human. # MAP1S pS812 is identified based on sequence homology. Red = phosphorylated serine, bold = RPXS consensus sequence.

E–H   *In vitro* kinase assays showing efficient MAP1S phosphorylation by CDKL5. 50 ng (40 nM) AS-CDKL5 phosphorylates 150 ng (50 nM) MAP1S very rapidly (E, F). In 30 min of incubation, 150 ng MAP1S is phosphorylated by small amounts of CDKL5 (G, H). Quantification of phosphorylated MAP1S is normalized to maximum intensity. *n* = 2 replicates.

I–L   Western blots of *in vitro* kinase assays showing loss of phosphorylation in phosphomutants ARHGEF2 S122A (I), EB2 S222A (J) and MAP1S S786/812A (K). Substrate levels are shown by Coomassie staining underneath. MAP1S phosphorylation is quantified in (L). Dunnett's multiple comparison: *n* = 3 replicates.

Data information: Thio-P, anti-thiophosphate ester antibody; n.s., not significant, *$P < 0.05$, ***$P < 0.001$, ****$P < 0.0001$, error bars are SEM. In panel (K), non-relevant lanes have been removed for simplicity (see accompanying Source Data for full blot).
Source data are available online for this figure.

Ser786 phosphopeptides specifically in AS-kinase samples and not in negative controls (Fig EV1, Appendix Figs S2 and S3). All phosphorylation sites contained an RPXS* consensus motif, where X is any amino acid and S* is the phosphorylated residue (Fig 1D). To validate these putative substrates, we performed *in vitro* kinase assays using purified AS-CDKL5 and purified substrates. We found that all three substrates are highly efficiently phosphorylated by CDKL5 in a time-dependent (Fig 1E and F, and Appendix Fig S4) and CDKL5 concentration-dependent manner (Fig 1G and H, and Appendix Fig S4). Next, to confirm the identified phosphorylation

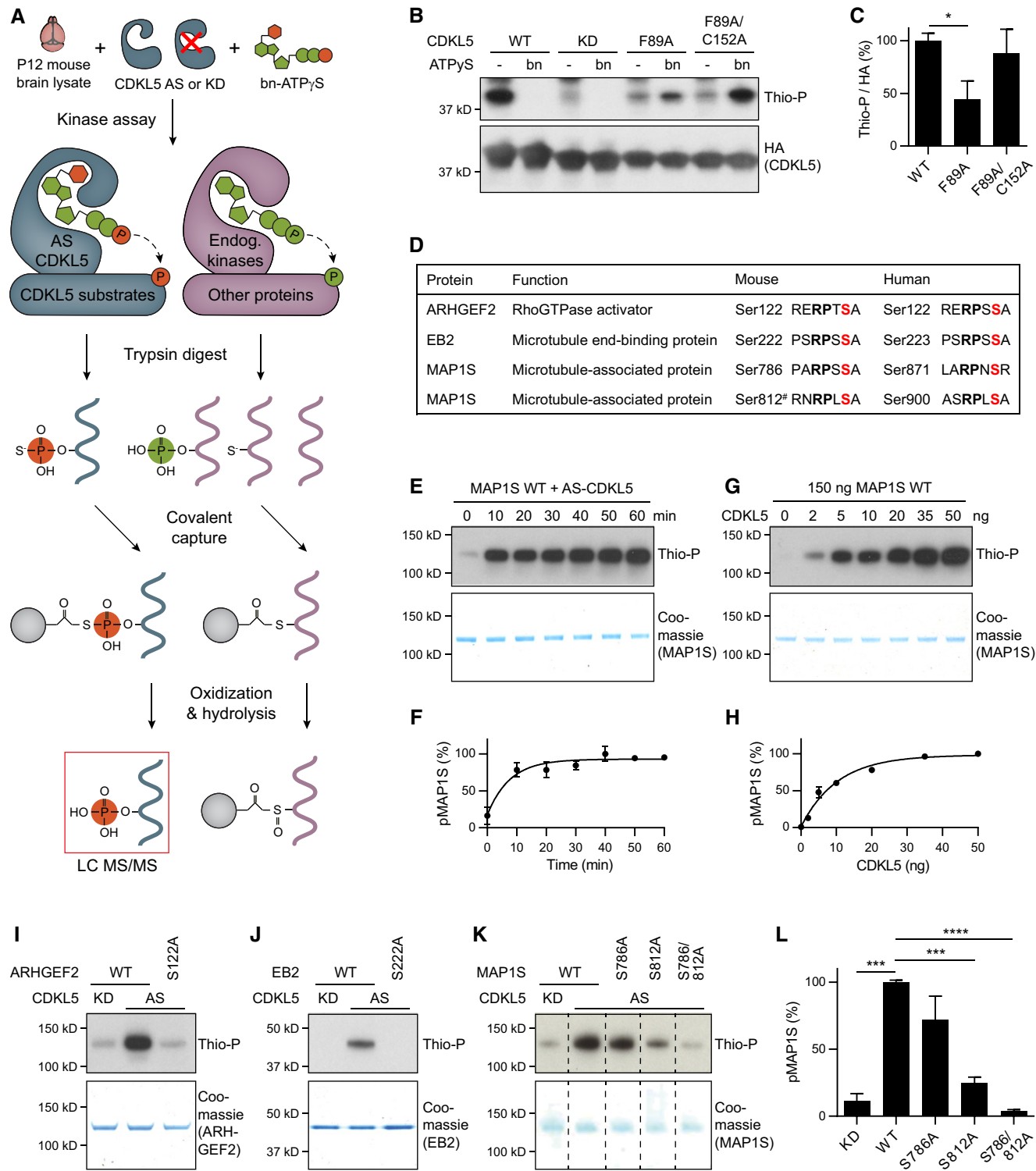

**Figure 1.**

sites, we generated phosphomutant substrates by mutating the Serines to Alanines and performed *in vitro* kinase assays. We found that ARHGEF2 S122A (Fig 1I) and EB2 S222A (Fig 1J) are not phosphorylated by CDKL5, confirming the mutated Serines as the sole phosphorylation sites on these substrates. In contrast, full-length MAP1S phosphorylation was reduced in MAP1S S786A mutant but not completely eliminated. Upon inspection of MAP1S protein sequence, we determined another putative phosphorylation site with the same RPXS consensus motif at Ser812. MAP1S phosphorylation was lost in double phosphomutant S786/812A, indicating that CDKL5 can phosphorylate MAP1S on both sites (Fig 1K and L). In summary, our chemical genetic screen identified three novel CDKL5 substrates on which four phosphorylation sites containing RPXS* motif were confirmed (Fig 1D). Interestingly, all substrates are microtubule-associated proteins, suggesting a role for CDKL5 in microtubule regulation.

## MAP1S and EB2 are physiological substrates of CDKL5 in brain

In order to study the phosphorylation events more closely, we generated rabbit polyclonal phosphospecific antibodies targeting the four mouse phosphorylation sites that we identified (Fig 1D). We expressed wild-type (WT) and phosphomutant substrates, ARHGEF2, EB2 and MAP1S, together with full-length wild-type or kinase-dead CDKL5 in HEK293 cells. We found that ARHGEF2 Ser122 and MAP1S Ser786 phosphorylation levels were increased with CDKL5 overexpression, demonstrating that these substrates can be phosphorylated by CDKL5 in cells, while EB2 Ser222 and MAP1S Ser812 sites were already highly phosphorylated endogenously (Fig EV2A–C). Importantly, for all four cases, phosphospecific antibodies recognized the WT but not the phosphomutant substrates, indicating their specificity for phosphorylated substrates.

Using these antibodies, we tested if the novel substrates are phosphorylated by CDKL5 *in vivo* in mice. CDKL5 protein was shown to be expressed in the cortex of wild-type (WT) mice and completely lost in CDKL5 knockout (KO) mice (Fig 2A). We found that endogenous EB2 Ser222 (pS222) and MAP1S Ser812 (pS812) phosphorylations are dramatically reduced in CDKL5 knockout mice cortex at P20 (Fig 2A and B). ARHGEF2 Ser122 and MAP1S Ser786 phosphospecific antibodies did not detect any specific signal, possibly due to insufficient sensitivity of these antibodies. These data indicated MAP1S and EB2 to be physiological substrates of CDKL5.

We also investigated the temporal and spatial distribution of CDKL5 activity. For this purpose, we examined the developmental time course of CDKL5 expression and of EB2 and MAP1S phosphorylation in mouse cortex. When comparing WT and KO cortical lysates from P4, P12, P20 and P50, we found that at all ages EB2 pS222 and MAP1S pS812 are highly dependent on CDKL5, as shown by absence of phosphorylations in CDKL5 KO (Figs 2C and D, and EV2D and E for full molecular range). Interestingly, while CDKL5 protein level increased postnatally and remained at its highest level in adult, CDKL5 activity towards EB2 pS222 and MAP1S pS812 was reduced significantly through 2nd to 3rd postnatal weeks. In order to assay phosphatase regulation of these substrates, we treated cultured neurons at younger (DIV11) or older (DIV22) developmental stages with PP1/PP2A type phosphatase inhibitor, okadaic acid (OA). We found that OA increased EB2 pS222 levels in both stages (Fig EV3A). The fold increase in EB2 pS222 was larger at DIV22 when compared to DIV11, indicating that PP1/PP2A activity may be higher in older stages than in younger neurons (Fig EV3B). However, even in the presence of OA the observed decrease in EB2 phosphorylation in older neurons remained, indicating that additional factors likely contribute to observed reduction in EB2 pS222 (Fig EV3C).

Next, we examined the cell-type specificity of CDKL5 activity by deleting CDKL5 in excitatory neurons using Nex-Cre mice (Goebbels *et al*, 2006) crossed with floxed CDKL5 mouse (Amendola *et al*, 2014). We observed that the vast majority of CDKL5 is found in excitatory neurons and that EB2 Ser222 and MAP1S Ser812 phosphorylations are significantly reduced at P8 in cortex of excitatory neuron-specific CDKL5 conditional knockouts (Fig 2E and F). In order to determine the subcellular distribution of phosphorylated CDKL5 substrates, we immunostained dissociated cortical neuron cultures with EB2 pS222. We found that total EB2 and EB2 pS222 were largely present in dendrites and EB2 pS222 staining was absent in CDKL5 KO cultures confirming its specificity (Fig 2G). In order to assess the specificity of EB2 pS222 staining, we knocked down endogenous EB2 using a previously reported shRNA sequence (Komarova *et al*, 2005). In contrast to scrambled shRNA control, EB2 shRNA caused a dramatic reduction in total EB2 and pS222 in dendrites of neurons expressing these plasmids along with GFP (Fig EV2F). Nuclear staining present with pS222 was non-specific background, as it was observed with total EB2 staining and it was not reduced by EB2 shRNA. These findings indicated that CDKL5 phosphorylates EB2 and MAP1S *in vivo* during early postnatal development and that

**Figure 2.  MAP1S and EB2 are physiological substrates of CDKL5 in mammalian brain.**

A, B   *In vivo* substrate phosphorylations are assessed by Western blotting of 5 WT (CDKL5+/Y) and 5 CDKL5 KO (CDKL5−/Y) mouse cortical lysates using phosphospecific antibodies. EB2 pS222 and MAP1S pS812 are dramatically reduced in KOs. Quantification of substrate phosphorylation is normalized for total protein level. Student's *t*-test: *n* = 5 animals per genotype.

C, D   Developmental expression of CDKL5 and levels of its substrate phosphorylations in P4–P50 cortex. CDKL5-dependent substrate phosphorylation is quantified by subtracting KO from WT levels for each age. *n* = 3 animals per age per genotype.

E, F   Expression of CDKL5 and levels of substrate phosphorylation in P8 Nex-Cre conditional KO cortex. CDKL5 protein expression is almost completely lost. Quantification of substrate phosphorylation is normalized for total protein level. Student's *t*-test: *n* = 3 animals per genotype.

G   CDKL5 WT and CDKL5 KO neurons in culture are co-stained with neuronal dendrite marker MAP2, EB2 and EB2 pS222. EB2 pS222 is lost in CDKL5 KO dendrites. Scale bar is 20 μm.

Data information: EB2 isoform 1/2 (37 kD) is quantified. ****$P < 0.0001$, error bars are SEM.
Source data are available online for this figure.

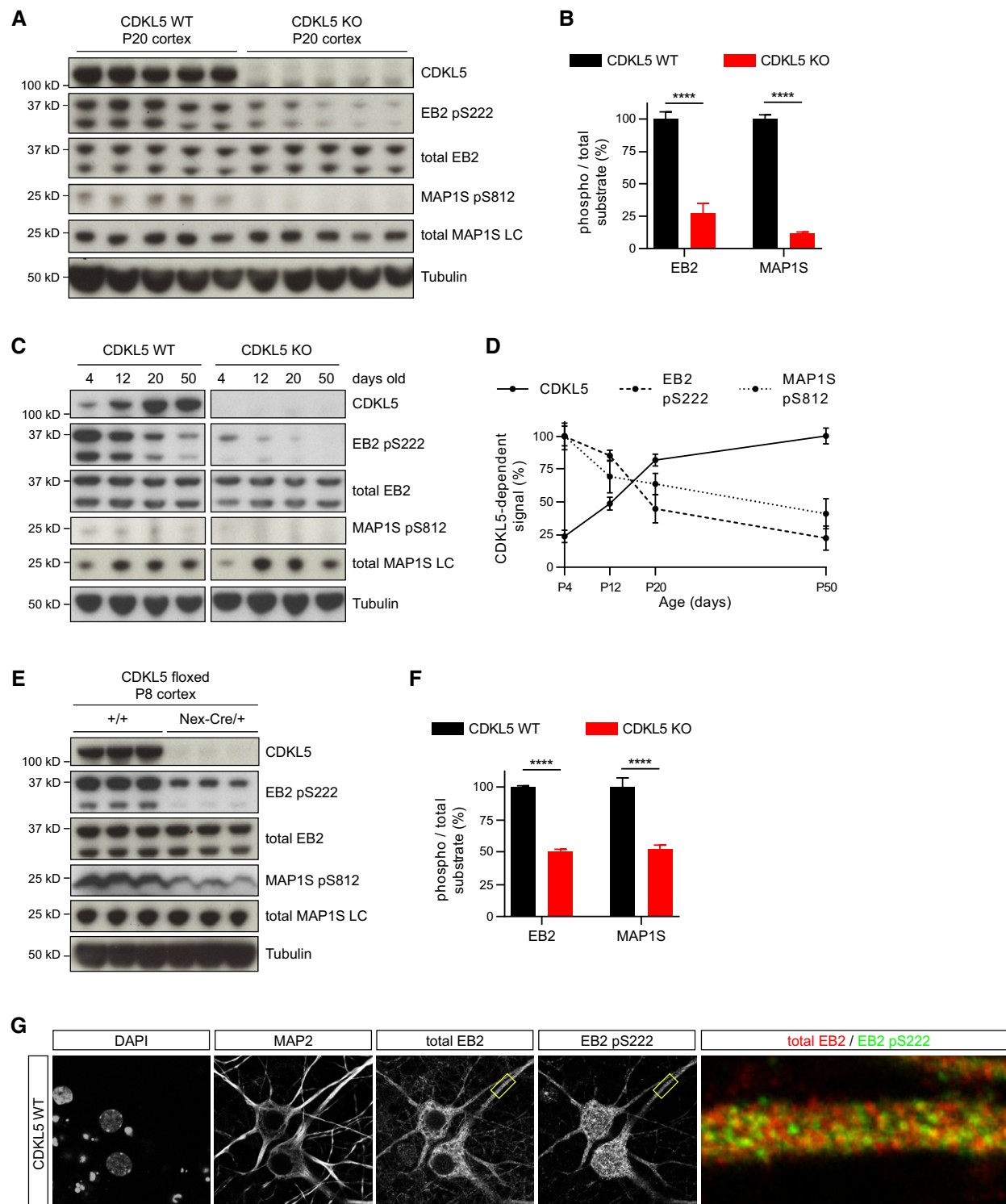

**Figure 2.**

CDKL5 is highly active in dendrites of pyramidal neurons in the cortex.

## EB2 Ser222 phosphorylation is suppressed by NMDA receptor activity

Neuronal activity and NMDA receptors are crucial in sculpting developing neuronal circuits and for activity-dependent plasticity in adults (Wong & Ghosh, 2002). We next tested if CDKL5 substrate phosphorylation was affected by NMDAR-dependent neuronal activity. We treated cortical neuronal cultures with 50 mM KCl for varying durations to induce neuronal depolarization. We found that phosphorylation of EB2 Ser222 was reduced to 50% of its initial levels within 10 min of KCl treatment (Fig 3A and B). This effect was due to depolarization and not because of altered osmolarity

(Appendix Fig S3C). Total CDKL5 levels were also reduced by 25% (Fig 3C). To test the role of NMDAR on EB2 pS222 regulation, we inhibited NMDAR with 100 µM AP5 during KCl treatment and found that AP5 largely rescued the KCl-mediated EB2 pS222 reduction (Fig 3A and B). Finally, we applied NMDA on cultured neurons and found that NMDA application reduced EB2 phosphorylation, supporting NMDAR-dependent regulation of CDKL5 substrate phosphorylation (Fig 3D and E). These data showed that CDKL5's phosphorylation of EB2 is inhibited by NMDAR activity, suggesting that CDKL5 contributes to activity-dependent circuit formation.

### CDKL5 loss leads to reduced microtubule dynamics via MAP1S

We reasoned that since CDKL5 phosphorylates microtubule-binding proteins, its loss could lead to disruption of neuronal

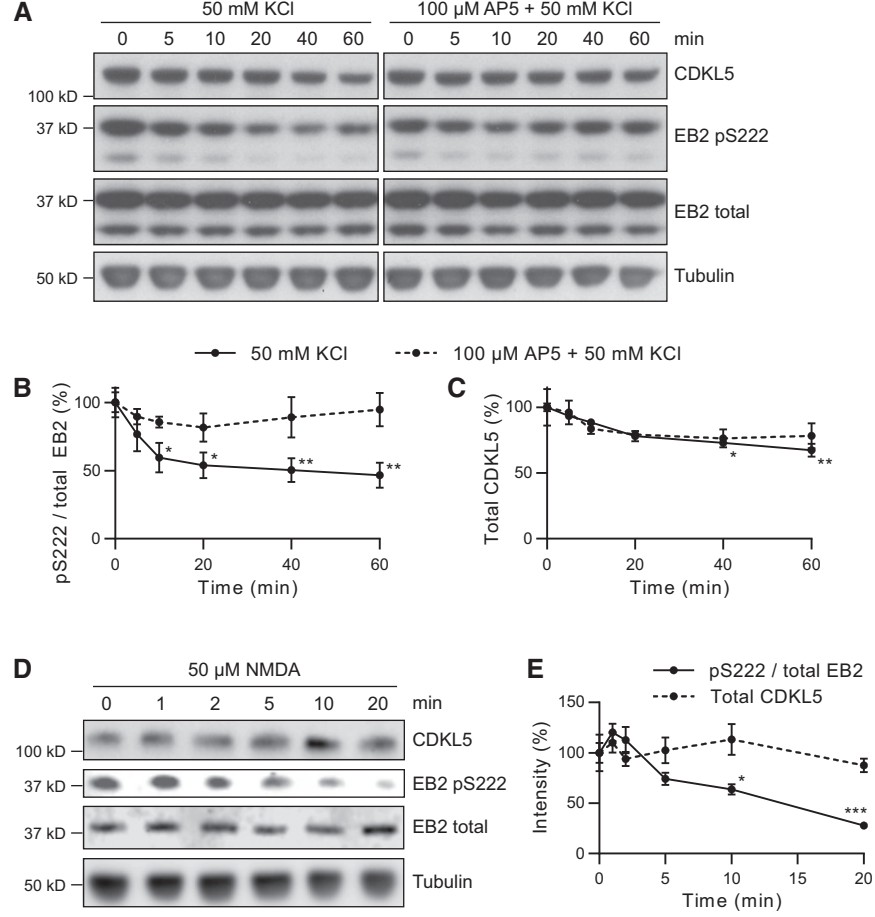

**Figure 3.  Phosphorylation of EB2 Ser222 is suppressed by NMDAR activity.**

A–C  Protein lysates were obtained from cortical neuronal cultures treated with 50 mM KCl applied directly to culture media for indicated durations. 100 µm AP5 was used to block NMDAR activity during KCl treatment. Quantification of phosphorylated EB2 (B) and total CDKL5 (C) is normalized to 0 min time point. EB2 phosphorylation is significantly decreased after 10 min of KCl treatment compared to 0 min. CDKL5 protein levels are slightly reduced after 40 min compared to 0 min. Dunnett's multiple comparison: *n* = 3 replicates.

D, E  Protein lysates were obtained from cortical neuronal cultures treated with 50 µM NMDA applied directly to culture media for indicated durations. Quantification of phosphorylated EB2 and total CDKL5 is normalized to 0 min time point. EB2 S222 phosphorylation is significantly reduced after 10 min compared to 0 min. Dunnett's multiple comparison: *n* = 3 replicates.

Data information: *$P < 0.05$, **$P < 0.01$, ***$P < 0.001$, error bars are SEM.
Source data are available online for this figure.

morphology. We tested this by examining CDKL5 KO mice in neuronal cultures and *in vivo*. In CDKL5 KO cortical primary neurons, we found a slight (10%) reduction in total dendrite length in cortical but not hippocampal cultures at 11 days *in vitro* (DIV; Fig EV4A–C). However, there was no significant alteration in basal dendrites of Thy1-YFP-expressing Layer V pyramidal neurons (Feng *et al*, 2000) in CDKL5 KO mice at P20 (Fig EV4D–F). In addition, axonal growth was not affected in CDKL5 KO neurons at DIV4 (Fig EV4G and H). These findings indicated that no major disruptions are present in dendrites of CDKL5 KO pyramidal neurons. While these results and others in literature (Zhou *et al*, 2017) did not provide compelling evidence for robust effects of CDKL5 on neuronal morphology, it prompted us to examine dendritic microtubules more closely.

Both EB2 and MAP1S are ubiquitously expressed in mammalian tissues and are implicated in regulating microtubule dynamics (Goldspink *et al*, 2013; Tegha-Dunghu *et al*, 2014). Given the predominant localization of EB2 pS222 on excitatory neuron dendrites, we decided to examine neuronal microtubule dynamics

in cultured cortical neuron dendrites. For this purpose, we imaged tdTomato-tagged EB3 in cortical neuron dendrites in dissociated cultures (Fig 4A, Movies EV1 and EV2). Cortical neuronal cultures were prepared from WT and CDKL5 KO littermates, transfected with EB3-tdTomato and live-imaged at DIV14. In agreement with previous reports (Stepanova *et al*, 2003; Ghiretti *et al*, 2016), we observed both anterograde (75%) and retrograde (25%) EB3 comets in dendrites, with no difference in their ratio between WT and CDKL5 KO (Fig EV5A). However, we found a robust increase in comet distance (2.06 μm in WT to 3.05 μm in CDKL5 KO) and comet lifetime (10.1 s in WT to 15 s in CDKL5 KO; Fig 4B and C) with no difference in comet velocity (Fig 4D). Results were similar when anterograde and retrograde comets were analysed separately (Fig EV5B). Next, we asked if EB2 or MAP1S could mediate CDKL5's effect on microtubule dynamics. We designed MAP1S shRNAs and used a known EB2 shRNA sequence (Komarova *et al*, 2005) to knock down these substrates in CDKL5 WT and KO cortical neurons (Fig EV5D). Both MAP1S shRNAs caused a significant decrease in comet lifetime in WT cortical neurons

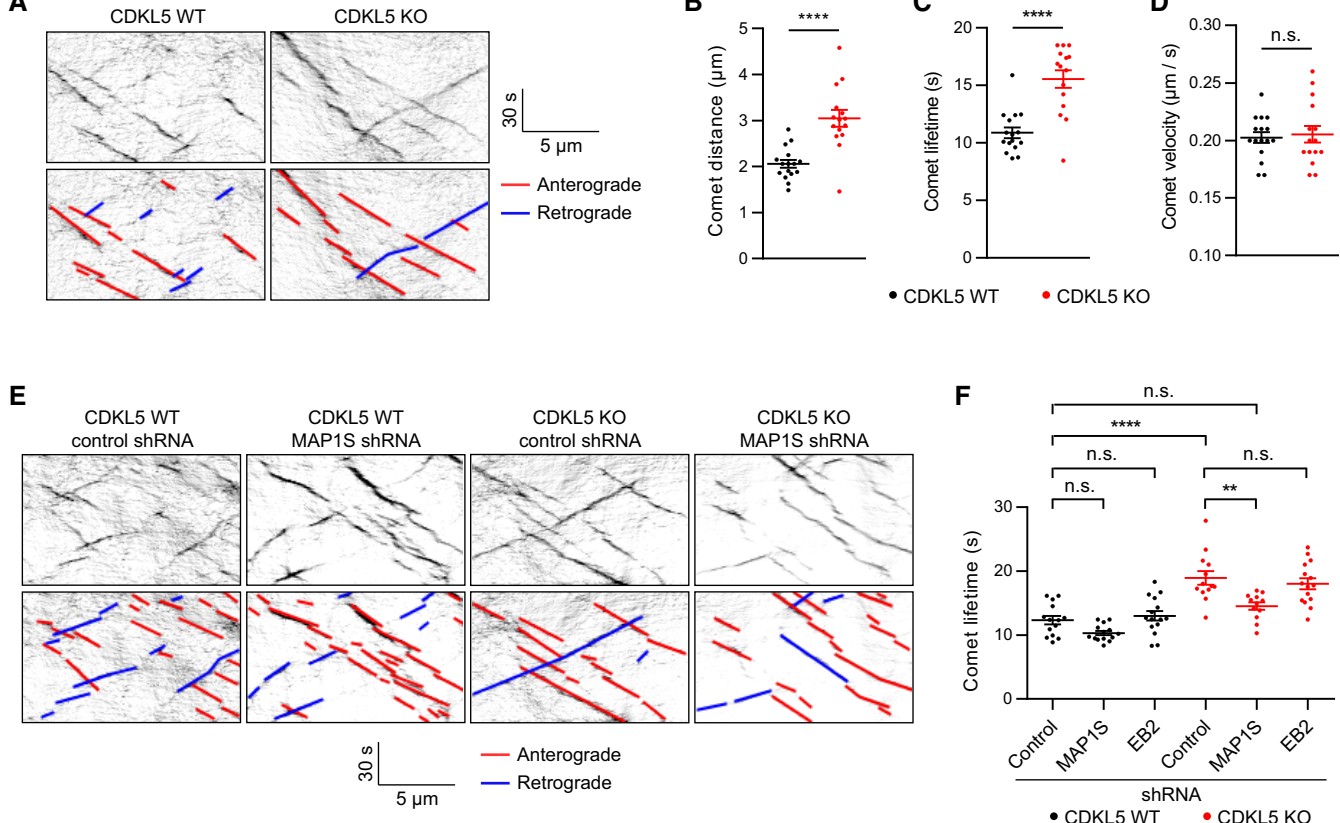

**Figure 4. Microtubule dynamics are reduced in CDKL5 KO dendrites via MAP1S.**

A   Representative kymographs of EB3-tdTomato in cortical neuron dendrites at DIV14.

B–D   In CDKL5 KOs (CDKL5$^-$/Y), comet distance (B) and lifetime (C) are increased when compared to WT (CDKL5$^+$/Y). Comet velocity (D) is unchanged. Student's *t*-test: *n* = 16/615 and 15/381 neurons/comets in WT and KO, respectively.

E   Representative kymographs of CDKL5 WT and KO neurons transfected with control shRNA and MAP1S shRNA.

F   Comet lifetime is higher in control shRNA expressing KOs than WTs. MAP1S shRNA expression reduces comet lifetime in KO neurons. EB2 shRNA-mediated knockdown does not alter microtubule dynamics in WT or KO. Tukey's multiple comparison: *n* = 12–15 neurons, 388–741 comets per condition.

Data information: n.s., not significant, **P < 0.01, ****P < 0.0001, error bars are SEM.

(Fig EV5E) but no significant changes in comet direction, density or velocity (Fig EV5F–H). Interestingly, MAP1S knockdown rescued the increase in comet lifetime in CDKL5 KO neurons, while EB2 knockdown had no effect (Fig 4E and F). These data indicated that CDKL5 positively regulates plus-end dynamic instability by phosphorylating MAP1S, causing shorter periods of plus-end growth.

### CDKL5 is necessary for efficient anterograde trafficking along dendrites

Microtubule plus tip dynamics and microtubule stability affect cargo trafficking along neuronal dendrites (Tas *et al*, 2017). Microtubule-based trafficking of cargo such as TrkB is important for neuronal function (Zweifel *et al*, 2005), and in dendrites, it is dependent on minus-end directed dynein and plus-end directed kinesin (Hirokawa *et al*, 2010; Ayloo *et al*, 2017). Therefore, we decided to analyse dendritic transport of BDNF/TrkB, a well-studied cargo for microtubule-based transport, and tested if trafficking of BDNF/TrkB carrying endosomal vesicles was affected in CDKL5 KO neurons. In TrkB-RFP-transfected cortical neuron dendrites, we could observe numerous TrkB-RFP endosomes moving in both anterograde and retrograde directions, as previously described (Ghiretti *et al*, 2016; Fig 5A, Movies EV3 and EV4). We analysed these using a particle tracking algorithm (Fig 5B, Movies EV5 and EV6). In CDKL5 KO neurons, we found no difference in the fraction of stationary

TrkB-RFP puncta (Fig 5C), the ratio of anterograde/retrograde runs (Fig 5D) or the velocity of the puncta, which was comparable to previous reports (Fig 5E). Interestingly, we observed a significant reduction in run lengths of anterograde TrkB puncta in CDKL5 KO dendrites (Fig 5F). These data showed that in CDKL5 deficiency anterograde cargo transport is compromised.

### CDKL5's phosphorylation of MAP1S light chain inhibits its microtubule binding

The finding that loss of MAP1S phosphorylation in CDKL5 KO neurons is rescued by MAP1S knockdown raised the possibility that CDKL5 negatively regulates MAP1S function. MAP1S is a member of MAP1 family of microtubule-associated proteins, which are proteolytically cleaved into heavy and light fragments that form a complex on microtubules. Ser786 and Ser812 phosphorylation sites are found on MAP1S light chain's (LC) microtubule-binding domain (Orban-Nemeth *et al*, 2005; Fig 6A). We tested if phosphorylation of these sites alters microtubule-binding affinity of MAP1S LC by over-expressing WT or kinase-dead CDKL5 with WT or S786/812A phosphomutant MAP1S in COS-7 cells. We found that MAP1S LC's robust localization to microtubules is strongly disrupted upon CDKL5 WT expression and this effect is dependent on the phosphorylation sites (Fig 6B–D). Furthermore, microtubule co-sedimentation assay showed that purified MAP1S LC binds polymerized microtubules (pellet) and that pre-phosphorylation of MAP1S LC

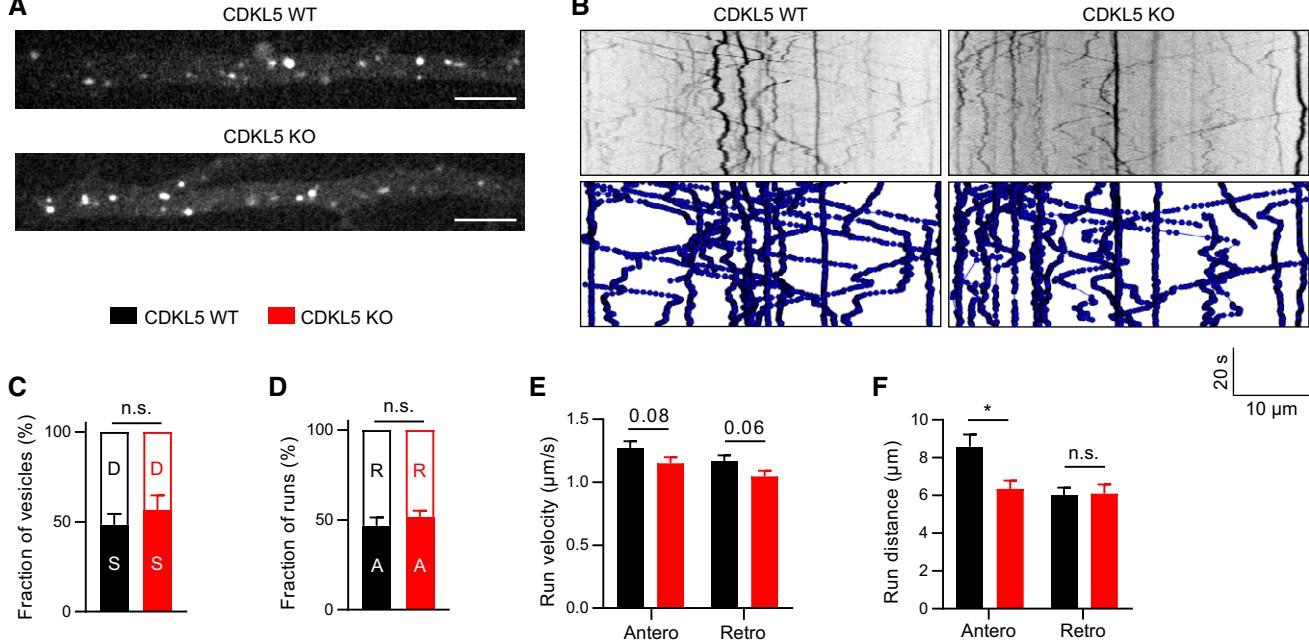

**Figure 5.  TrkB trafficking is impaired in CDKL5 KO dendrites.**

A, B    Representative stills (A) and kymographs (B) of TrkB-RFP overexpressed in dendrites of WT and CDKL5 KO DIV14 cortical neurons. Scale bar is 5 μm.

C, D    No significant change is observed in the ratio of dynamic vs. stationary vesicles (C) or anterograde vs. retrograde runs (D). S, stationary; D, dynamic; A, anterograde; R, retrograde. Vesicles with a net distance > 5 μm were considered dynamic.

E, F    Both anterograde and retrograde runs show a trend towards reduced velocity in CDKL5 KO neurons (E). Anterograde run lengths are significantly reduced in KO compared to WT neurons (F).

Data information: Student's *t*-test: *n* = 4 cells/121–138 runs per condition. n.s., not significant, *P < 0.05, error bars are SEM.

     

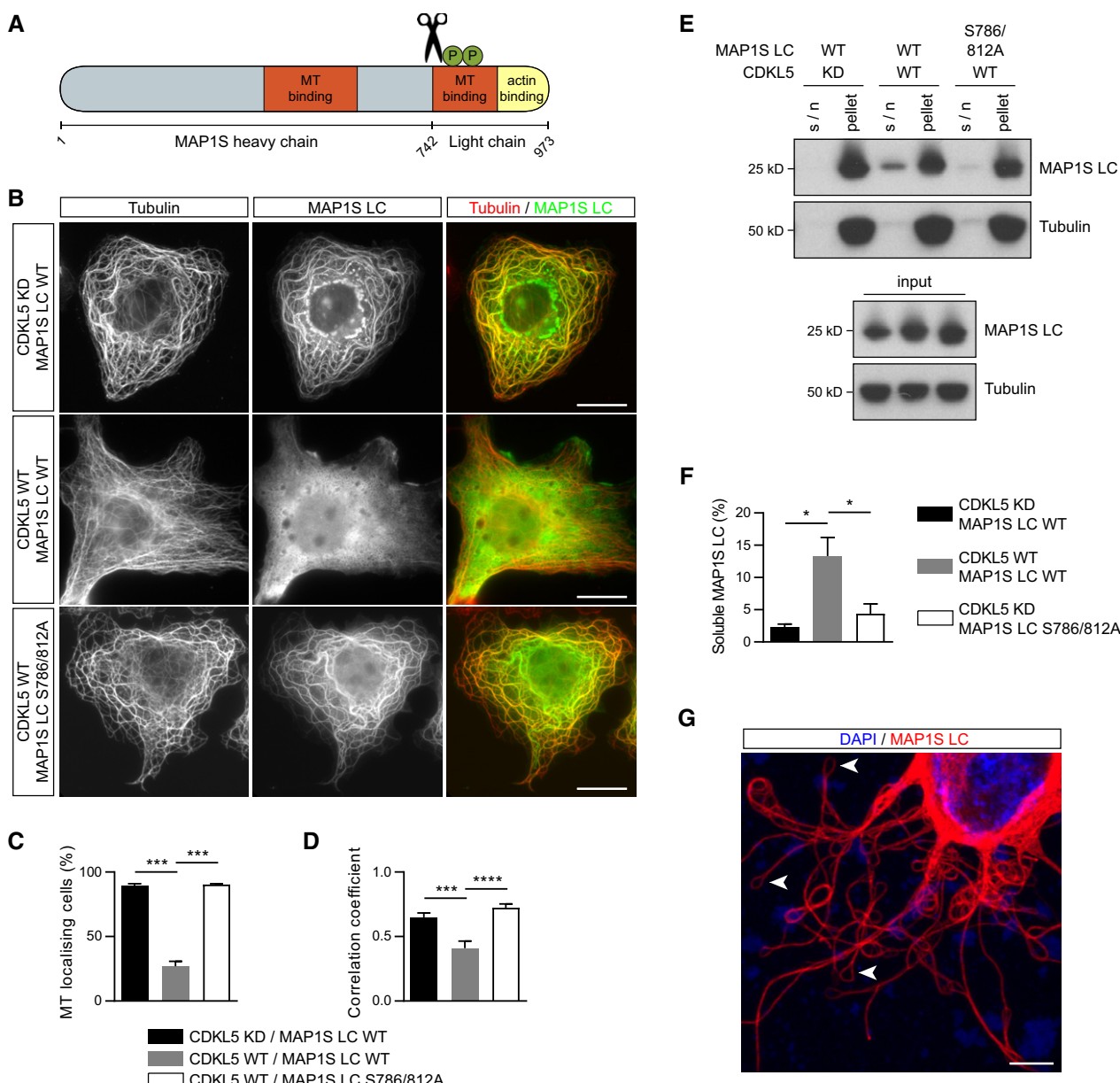

**Figure 6. MAP1S phosphorylation by CDKL5 impairs its microtubule binding.**

A   Schematic representation of MAP1S protein domains. Phosphorylation sites S786 and S812 (green Ps) are on the microtubule-binding domain of MAP1S LC. Scissors indicate proteolytic cleavage.

B   COS-7 cells expressing HA-MAP1S LC, Flag-CDKL5 WT/kinase-dead (KD) is shown with endogenous α-tubulin staining. MAP1S does not localize to microtubules when phosphorylated by CDKL5. Scale bar is 20 μm.

C, D   The percentage of cells where α-tubulin and MAP1S LC are colocalized (C) and the quantification of MAP1S LC and α-tubulin colocalization by Pearson correlation coefficient (D). Tukey's multiple comparison: *n* = 22–24 cells per condition.

E, F   Taxol-stabilized microtubule co-sedimentation assay shows direct binding of MAP1S LC to microtubules is reduced upon prior phosphorylation by CDKL5. Quantification of soluble MAP1S in the supernatant (s/n) fraction is normalized to the MT-bound pellet fraction. 3:1 MT:MAP1S LC ratio used here. Tukey's multiple comparison: *n* = 3 technical repeats.

G   HA-MAP1S LC is overexpressed and stained with anti-HA antibody in cultured cortical neurons (red). Numerous loops at the dendrites' tips are observed (arrowheads). Scale bar is 5 μm.

Data information: *$P < 0.05$, ***$P < 0.001$, ****$P < 0.0001$, error bars are SEM.
Source data are available online for this figure.

with CDKL5 highly increased soluble MAP1S (supernatant; Fig 6E and F). MAP1S LC phosphorylation is likely to be saturated as pS786 and pS812 antibodies were not enriched in the soluble

fraction (Appendix Fig S5A and B). *In vitro* MAP1S LC microtubule-binding affinity is therefore also dependent on other factors such as MAP1S LC:Tubulin ratio. A putative phosphomimetic MAP1S LC

mutant did not act like phosphorylated MAP1S LC and bound to microtubules in the microtubule co-sedimentation assay and COS-7 cells (Appendix Fig S5C and D). When MAP1S LC S786/812A was overexpressed in neurons, microtubules formed loops at the tips of the dendrites, possibly due to stabilization or continued growth (Fig 6G). These data support that binding of MAP1S LC to microtubules is directly regulated by CDKL5-dependent Ser786 and Ser812 phosphorylation.

### EB2 phosphorylation is lost in patient iPSC-derived neurons

In order to test whether the phosphorylation of novel substrates was also altered in human subjects with CDD, we collected fibroblasts from three patients: p.R59X, female; p.R59X, male; D135_F154del, male and three controls. The R59X mutation leads to a premature stop codon and loss of CDKL5 protein by nonsense mediated decay,

while D135_F154del leads to a truncated protein missing exon 7. The fibroblasts were reprogrammed into iPSCs (Appendix Fig S6) and then differentiated into neurons. We found that in patient-derived neurons EB2 pS222 levels are reduced to 20% of control phosphorylation level (Fig 7A and B). EB2 pS222 immunostaining was also greatly reduced in patient-derived neuronal dendrites (Fig 7C). These data show that novel CDKL5 substrates identified in mice are conserved in humans and that the loss of CDKL5 in humans promotes significant reductions in EB2 phosphorylation levels.

In conclusion, in this study we report novel physiological CDKL5 substrates that are altered in neurons from CDD patients. Our molecular and cellular analysis of CDKL5 and MAP1S function suggests that MAP1S binding to microtubules causes longer plus-end growth and less plus-end dynamics. Altered microtubule architecture in turn reduces efficient cargo trafficking (Fig 8).

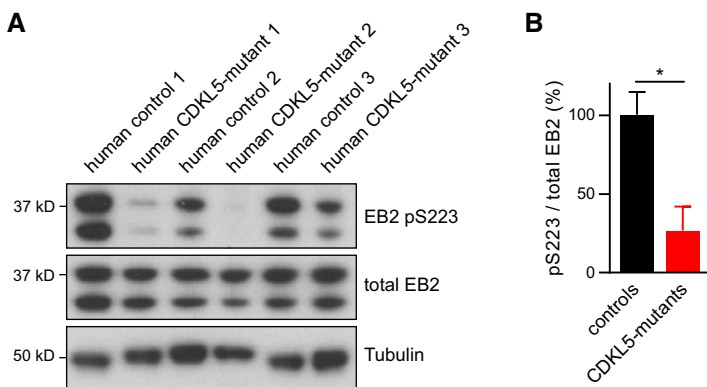

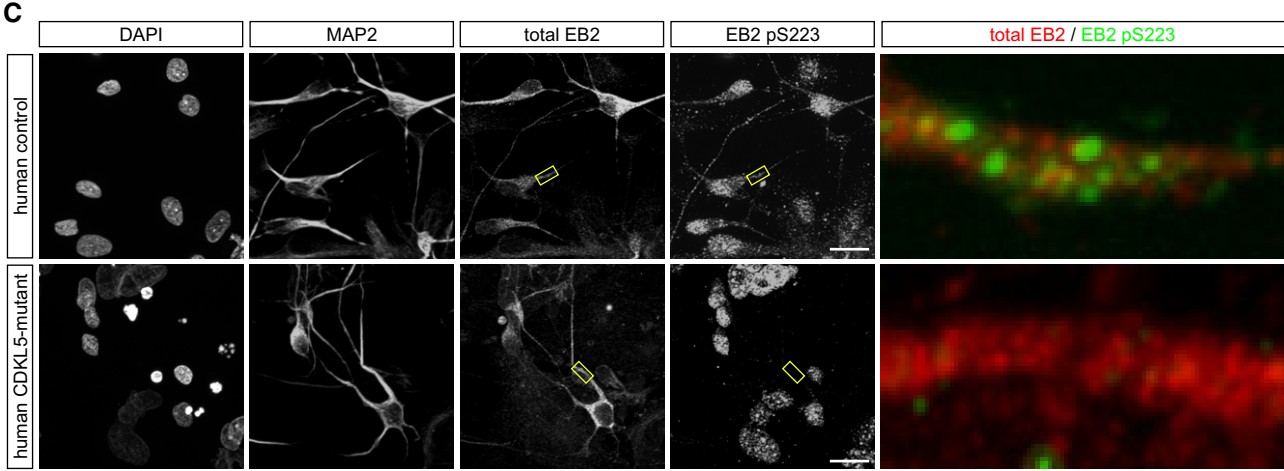

**Figure 7. EB2 S223 phosphorylation is lost in CDD patient-derived neurons.**

A, B Western blots for total EB2, pS223 and tubulin using protein lysates obtained from neurons derived from three patients with CDD and their related control. 1 = p.D135_F154del, male; 2 = p.R59X, male; 3 = p.R59X, female. Quantification of EB2 phosphorylation is normalized for total protein level. Student's t-test: n = 3 patients/controls. *P < 0.05, error bars are SEM.

C Representative images of human iPSC-derived neurons co-stained with the neuronal dendrite marker MAP2, EB2 and EB2 pS223. EB2 phosphorylation is present in dendrites of control neurons but not in CDD neurons. Enlarged images of the boxed areas are shown on the right panel. Scale bar is 20 μm.

Data information: human EB2 pS223 is mouse/rat EB2 pS222.
Source data are available online for this figure.

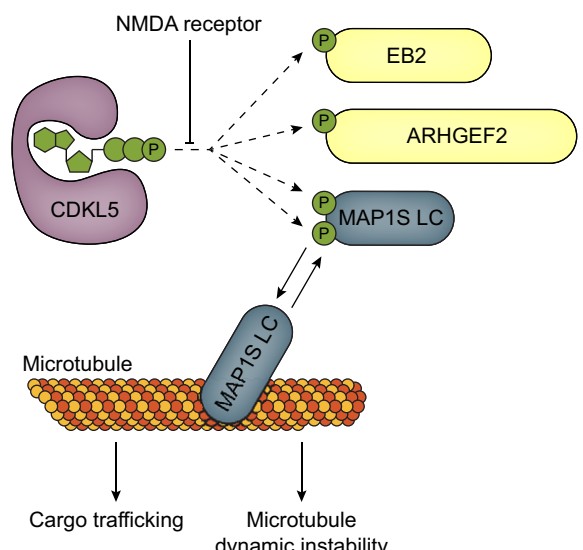

**Figure 8. Summary schematic of novel CDKL5 substrates and functions in microtubule dynamics.**

CDKL5 phosphorylates microtubule-associated proteins EB2, MAP1S and ARHGEF2 at their RPXS motifs. Double phosphorylation of the MAP1S LC microtubule-binding region decreases MAP1S affinity for microtubules. Loss of these MAP1S phosphorylations leads to impaired microtubule dynamic instability in CDKL5 KO neurons. Altered microtubule stability in CDKL5 KO neurons is also likely to be responsible for reduced cargo trafficking.

## Discussion

### Novel CDKL5 substrates

Identifying CDKL5 substrates has been a major target in CDKL5 deficiency disorder research in the last 10 years. Owing to the clinical resemblance of CDD to Rett syndrome, initially methyl-CpG-binding protein 2 (MECP2) was suggested to be a substrate (Mari *et al*, 2005). However, subsequent studies showed that MECP2 and DNA-Methyl transferase 1, another putative substrate identified as a CDKL5 binding partner (Kameshita *et al*, 2008), were weak *in vitro* substrates (Lin *et al*, 2005; Sekiguchi *et al*, 2013). Amphiphysin-1, a presynaptic protein, was identified in substrate screening using wild-type CDKL5 to phosphorylate cellular protein fractions (Sekiguchi *et al*, 2013). While AMPH1 Ser293 is an efficient *in vitro* substrate that contains the RPXS* consensus (Katayama *et al*, 2015), CDKL5-dependent regulation of AMPH1 remains to be demonstrated *in vivo*. Another study showed that Netrin G1 ligand (NGL-1) Ser631 phosphorylation is mediated by CDKL5. However, NGL-1 does not contain the RPXS* motif and it remains to be clarified if it is a direct target of CDKL5, e.g. with *in vitro* kinase assays (Ricciardi *et al*, 2012). The reduction in NGL-1 Ser631 phosphorylation in CDKL5 mutant cells is much less pronounced than the reductions in phosphorylation of EB2 and MAP1S shown here, suggesting that NGL-1 may be indirectly affected. Recently, a phospho-antibody array-based substrate screen reported histone deacetylase HDAC4 Ser632 as a putative CDKL5 phosphorylation site (Trazzi *et al*, 2016). Ser632 does not contain the CDKL5 consensus, and direct confirmation of this phosphorylation site by *in vitro* kinase assays remains to be done. The novel substrates we report here are direct

CDKL5 phosphorylation sites *in vitro* and MAP1S Ser812 and EB2 Ser222 are *in vivo* phosphorylation targets of CDKL5. Phosphorylation of these sites is not compensated by other kinases in the mouse brain. Our study reports three *bona fide* CDKL5 substrates and highlights the use of AS-kinases in determining physiological substrates.

Phosphorylated RPXS* motif is present in more than 1,000 reported phosphorylation sites (phosphosite.org). Therefore, many more CDKL5 substrates could be present and these remain to be discovered and validated.

### Cell type and subcellular localization of CDKL5 activity

Our results show that the vast majority of CDKL5 is found in pyramidal neurons. But incomplete reduction in EB2 pSer222 in conditional knockouts when compared to full knockout mice indicates that other cell type such as inhibitory neurons may also express low levels of CDKL5 and phosphorylate its substrates. We did not find any nuclear substrates of CDKL5, contrary to previous reports that indicate its nuclear roles (Rusconi *et al*, 2008; Ricciardi *et al*, 2009).

### CDKL5's regulation by neuronal activity

CDKL5's activity, as measured by phosphorylation of its substrates EB2 and MAP1S, is higher at earlier postnatal stages and reflects an early developmental role consistent with the early onset of CDD. Behavioural analysis of full or conditional CDKL5 KO in excitatory pyramidal neurons in cortex and hippocampus showed that CDKL5 activity is required for learning and memory (Wang *et al*, 2012; Tang *et al*, 2017). Corroborating its relevance in adult plasticity, CDKL5 substrate phosphorylations are present in adult, albeit at lower levels. We found that NMDA receptor activation causes a rapid reduction in EB2 pS222. It is possible that increased NMDA receptor activation upon maturation of synapses in adult mice inhibits baseline levels of EB2 phosphorylation. Our results also agree with previous work showing that extrasynaptic NMDA receptor stimulation negatively regulates CDKL5 levels in cultures (La Montanara *et al*, 2015). NMDAR signalling, phosphatase activity and developmental regulation of putative co-activators are among possible factors that could reduce these phosphorylations in adult. The reduction in CDKL5 phosphotargets with activity would be predicted to reduce dynamic instability, consistent with the NMDA-dependent loss of dynamic plus ends (Kapitein *et al*, 2011).

### CDKL5 as a regulator of microtubule dynamics and trafficking

Several families of microtubule-associated proteins regulate microtubule dynamics (Akhmanova & Steinmetz, 2015). Among these, MAP1 family members MAP1A, MAP1B and MAP1S are cleaved to form heavy and light chains, which can stabilize microtubules (Halpain & Dehmelt, 2006). In this study, we showed that phosphorylation dissociates MAP1S from microtubules, similar to the phosphoregulation of microtubule-binding protein Tau (Wang & Mandelkow, 2016). We also presented evidence that CDKL5 KO neurons have altered microtubule dynamics and this effect is rescued upon reducing MAP1S levels by shRNA. These results indicate that the more stable plus-end growth observed in CDKL5 KO neurons is dependent on MAP1S levels. Our results agree with the findings in HeLa cells where shRNA-mediated knockdown of MAP1S causes

shorter lived of EB3-EGFP comets (Tegha-Dunghu *et al*, 2014). In addition, MAP1S shRNA reduces microtubule acetylation globally indicating a reduction is stable microtubule population (Tegha-Dunghu *et al*, 2014). In support of MAP1S's role in microtubule stability, MAP1S light chain overexpression in COS-7 cells leads to highly stabilized microtubules, which are resistant to microtubule depolymerizing agents such as colchicine (Orban-Nemeth *et al*, 2005).

We investigated TrkB as a model cargo protein and found that its anterograde run length is reduced in dendrites. This would indicate that trafficking to distal dendritic regions could potentially be affected in CDKL5 KO. Different cargo proteins may be affected differently in CDKL5 KO animals, and additional experiments are needed to determine the endogenous cargo proteins that may be affected.

Despite its role in cytoskeletal and cargo transport regulation, we did not find major differences in dendrite morphologies of CDKL5 KO mice in culture or *in vivo*. Our results are in agreement with reports where subtle reductions in dendrite length were observed in CDLK5-deficient pyramidal neurons from CDKL5 KO mice (Tang *et al*, 2017; Zhou *et al*, 2017), but contrast with major dendritic alterations (Amendola *et al*, 2014). Modest reductions in dendrite arborization could be caused by compromised anterograde trafficking in CDKL5 KO dendrites. Alterations in tubulin posttranslational modifications could also affect selective molecular motor types or microtubule orientation (Tas *et al*, 2017). The roles of CDKL5 regulation of EB2 and ARHGEF2 remain to be studied to achieving a comprehensive understanding of CDKL5's influences on cytoskeleton.

### CDKL5 substrates in humans

EB2 pS222 antibody reliably reports CDKL5 activity in mice and in humans. Neither CDKL5 KO mice nor CDD patients EB2 phosphorylation is compensated by other kinases, making EB2 Ser222 phosphorylation an ideal read-out for CDKL5 activity and a potential molecular biomarker for pre-clinical or clinical studies. Our findings reveal a novel signalling mechanism for activity-dependent microtubule dynamics and cargo trafficking.

# Materials and Methods

### Animals

All mouse handling was performed according to the regulations of the Animal (Scientific Procedures) Act 1986. Animal studies and breeding were approved by the Francis Crick Institute ethical committee and performed under U.K. Home Office project licence (PPL 70/7771). Mice were housed in an animal facility of the Francis Crick Institute on an alternating 12-h light–dark cycle and provided with water and food *ad libitum*. All mouse lines were frequently backcrossed into C57BL/6J genetic background and displayed normal health and weight. CDKL5 KO and cKO mice were a kind gift from Cornelius Gross (Amendola *et al*, 2014). Heterozygous females and wild-type males were used to maintain CDKL5 KO mouse line. Brains were harvested at P4–50 of only male offspring to ensure CDKL5 WT and hemizygous KO littermates. For analysis of *in vivo* dendrite morphology at P20-50, Thy1-YFP or Thy1-GFP

expressing males were used for breeding with heterozygous CDKL5 females to introduce Thy1-YFP or Thy1-GFP transgene in the experimental male offspring. Targeted CDKL5 deletion in excitatory neurons was achieved by crossing homozygous floxed CDKL5 females with heterozygous Nex-Cre males. Brains were harvested at P8 of CDKL5 hemizygous floxed males expressing Nex-Cre and Cre-negative littermates as controls.

### Cell cultures

HEK293T and COS-7 cells were maintained in Dulbecco's modified Eagle medium (Gibco) containing 10% foetal bovine serum (Biosera) and penicillin/streptomycin at 37°C with 5% $CO_2$. Both lines had the species confirmed by Multiplex PCR. HEK293T was authenticated by STR profiling. Sf9 insect cells were cultured in suspension at 28°C in SF900 II serum-free medium (Invitrogen).

Mouse cortical and hippocampal neurons were cultured from individual male E16.5 embryos from a heterozygous CDKL5 mother. $CDKL5^+$/Y or -/Y genotype of embryos was determined afterwards. Neurons were plated at a density of 300,000 cells per 18 mm glass coverslips (Fisher) or 1,000,000 cells per 35 mm glass-bottom dishes (Mattek) coated with 60 μg/ml poly-D-lysine (Sigma) and 2.5 μg/ml laminin (Sigma). Neurons were plated with minimal essential medium (Gibco) containing 10% foetal bovine serum (Biosera), 0.5% dextrose, 0.11 mg/ml sodium pyruvate (Gibco) and 2 mM glutamax (Gibco). After 4 h, cultures were transferred to neurobasal medium (dye-free for glass-bottom dishes, Gibco) containing B27 (Gibco), 0.5 mM glutamax (Gibco), 12.5 μM glutamate, penicillin/streptomycin and ciprofloxacin. 1/3$^{rd}$ of media was replaced with fresh media every 3–4 days.

### Generation of iPSCs

Skin biopsies from three subjects diagnosed with CDKL5 deficiency disorder (CDD) and their first-degree sex-matched related control (parent) were kindly donated through a collaboration with the "International Foundation for CDKL5 Research" (p.R59X, female; p.R59X, male; D135_F154del, male). The skin fibroblasts were reprogrammed into iPSCs at the stem cell core at Sanford-Burnham Medical Research Institute. Episomal vectors carrying the reprogramming factors OCT3/4, SOX2, KLF4, LIN28, L-MYC and p53 shRNA were used as previously described (Okita *et al* 2011). iPSC colonies were cultured using mTeSR1 (Stem Cell Technologies) and expanded into matrigel (BD Biosciences)-coated plates. Written informed consent was obtained from all subjects, and this study was approved by the UCSD Human Research Protection Program Committee (IRB/ESCRO protocol #141223ZF).

### Generation of human CDD neural cultures

Neural differentiation was performed as described elsewhere (Thomas *et al*, 2017). Briefly, iPSC media was replaced with DMEM/F12 (Corning Cellgro) with 1× HEPES, 1× penicillin-streptomycin, 1× Glutamax (Life Technologies) and 1× N2 NeuroPlex (Gemini Bio-products), supplemented with 1 mM dorsomorphin (Tocris) and 10 mM SB431542 (StemGent). Next, colonies were kept in suspension for 7 days to form embryoid bodies (EBs), which were plated onto matrigel-coated plates and cultured using DMEM/

F12 with 1× HEPES, 1× penicillin-streptomycin, Glutamax, 0.5× N2 NeuroPlex and 1× Gem21 NeuroPlex (Gemini Bio-products), supplemented with 20 ng/ml bFGF (Life Technologies). Neural rosettes were manually collected, gently dissociated with accutase (Stem Cell Technologies), and the neural progenitor cells (NPCs) were replated on poly-L-ornithine/laminin-coated plates. Next, the NPCs were differentiated into neurons by adding of 5 μM ROCK inhibitor (Tocris) for 48 h and bFGF withdraw for 6 weeks.

## DNA constructs and shRNAs

N-terminal FLAG-tagged full-length CDKL5 107 (NM_001323289.1) in pcDNA3 was a kind gift from Charlotte Kilstrup-Nielsen at the University of Insubria. Human CDKL5[1–352] was cloned into pRK5 mammalian expression vector with N-terminal HA-tag and pFastBac HT insect cell expression vector with N-terminal His$_6$-tag. CDKL5 kinase-dead K42A and analog-sensitive mutations F89A and C152A were generated by site-directed mutagenesis. Mouse ARHGEF2 (NM_008487.3), mouse EB2 (NM_153058.4), rat MAP1S (NM_001106070.1) full length and light chain (LC, 754–972 aa) were cloned into pTriEx-6 (Novagen) to produce an N-terminal Strep(II) fusion. Mouse EB2 and rat MAP1S LC were also cloned into pRK5 mammalian expression vector with N-terminal HA-tag. Phosphomutant mutations ARHGEF2 S122A, EB2 S222A and MAP1S S786A, S812A and S786/812A were generated by site-directed mutagenesis. Phosphomimetic MAP1S LC S786/812D was generated by site-directed mutagenesis. EB3-tdTomato was a gift from Erik Dent (Addgene plasmid # 50708). TrkB-RFP was a kind gift from Erika Holzbaur. All constructs were confirmed by DNA sequencing.

shRNA target sequences on MAP1S were 22 base pairs long and selected via http://katahdin.cshl.org/: #1 5′-ACCTCACTGTGTCCT GTCCAAC-3′ and #2 5′-AGGCTTTACAGTGCTGGTGAAC-3′. EB2 shRNA target sequence was 19 base pairs long (Komarova *et al*, 2005): 5′-GATGAATGTTGATAAGGTA-3′. Nineteen base pair scrambled shRNA target sequence was used as a control: 5′-AGACCCAAG-GATTAGAAGG-3′. Hairpins targeting these sequences were cloned in pLentiLox 3.7 which expresses EGFP via a separate promoter in addition to expressing shRNA via a U6 promoter. Empty pLentiLox 3.7 vector was used as a cell fill to visualize neurons and referred to as GFP. shRNA efficiency was validated by co-expressing 0.5 μg HA-EB2 or HA-MAP1S LC with 1.5 μg corresponding shRNA in HEK293T cells for 48 h using Xtremegene 9 (Roche) according to manufacturer's instructions. Protein levels were detected by anti-HA antibody on Western blots.

## Western blotting

Mouse cortices were solubilized directly in sample buffer containing 0.1 M DTT. Lysates were sonicated briefly three times and centrifuged at 20,000 *g* for 15 min. All protein samples were denatured at 95°C for 5 min and ran on NuPAGE 4–12% Bis–Tris polyacrylamide gels (Invitrogen). Proteins transferred to Immobilon PVDF membrane (Millipore) were blocked using 5% milk in TBST. Primary antibodies incubated at 4°C o/n and HRP-conjugated secondary antibodies (Jackson Immunoresearch) incubated at RT for 2 h. Signal was detected using enhanced chemiluminescence (Pierce). Quantification of Western blots was manually performed using Fiji Gel Analyzer. Phosphorylations are measured relative to the total protein level, except for the developmental expression, where signals were normalized to tubulin. Intensities were normalized to the highest average value depicted as 100%.

The following primary antibodies were used for Western blotting: rat anti-HA (1:2,000, Roche 11 867 423 001), rabbit anti-thiophosphate ester (1:30,000, Abcam ab92570), rat anti-FLAG (1:2,000, Thermo Scientific MA1-142), mouse anti-Strep(II) (1:1,000, IBA 2-1507-001), mouse anti-αTubulin (1:50,000, Sigma T9026), rabbit anti-CDKL5 (1:1,000, Abcam ab22453), rabbit anti-CDKL5 (1:1,000, Sigma HPA002847), rat anti-EB2 (1:1,000, Abcam ab45767), rabbit anti-MAP1S LC (1:1,000, Sigma HPA050934).

## Phosphospecific antibodies

Rabbit polyclonal phosphospecific antibodies were raised against the following phosphorylated (*) peptides by Covalab: TTRERPT-S*AIY (mouse ARHGEF2 pS122), PGSTPSRPSS*AKRA (mouse EB2 pS222), RKAPARPSS*ASAT (mouse MAP1S pS786), AGDRNRPL-S*ARSE (mouse MAP1S pS812). The peptide sequence for EB2 pS222 also matches 100% with rat and human sequences. Final bleeds of immunized New Zealand White rabbits are purified with affinity purification by Covalab: The immune serum is loaded onto a column with the control peptide coupled to agarose beads, thus retaining unmodified peptide-specific antibodies. The flow-through is then loaded onto a column with the modified peptide coupled to agarose beads, thus retaining the modified peptide-specific antibodies. After elution, the eluate is assayed by ELISA against both peptides to control its immunoreactivity and its specificity against the modification. Phosphospecific antibodies were used at the following dilutions: rabbit anti-ARHGEF2 pS122 (1:2,000), rabbit anti-EB2 pS222 (1:2,000), rabbit anti-MAP1S pS786 (1:2,000), rabbit anti-MAP1S pS812 (overexpressed 1:2,000, endogenous 1:500).

## Protein purification

HA-CDKL5[1–352] and Strep(II)-ARHGEF2, Strep(II)-EB2 and Strep(II)-MAP1S were overexpressed in HEK293T cells for 48 h using Xtremegene 9 (Roche) according to manufacturer's instructions. HA-tagged and Strep(II)-tagged proteins were purified by anti-HA affinity matrix (Roche) and Strep-tactin resin (IBA), respectively. Lysis buffer contained 20 mM Tris–HCl pH 8.0, 150 mM NaCl, 1% NP-40, 10% glycerol, 1× protease inhibitor cocktail (Roche) and 1:100 phosphatase inhibitor cocktail III (Roche). Lysis was achieved by incubation on ice for 30 min. Cells were pipetted up and down several times before centrifugation at 20,000 *g* for 15 min at 4°C. Supernatant containing solubilized HA-tagged proteins was precleared using IgG-Sepharose (GE Healthcare) for 30 min at 4°C and immunoprecipitated with HA affinity matrix for 2 h at 4°C. Supernatant containing solubilized Strep(II)-tagged proteins was purified at 4°C on columns with Strep-tactin resin. Proteins bound on HA-matrix and Strep-tactin resin were washed three times. HA-bound CDKL5[1–352] was washed an additional two times with kinase buffer containing 20 mM Tris–HCl pH 7.5, 10 mM MgCl2, 1× protease inhibitor cocktail (Roche) and 0.5 μM okadaic acid. Strep (II)-tagged proteins were eluted by 50 mM desthiobiotin, and protein concentration was measured by comparison to BSA standards.

His$_6$-CDKL5$^{1–352}$ KD, His$_6$-CDKL5$^{1–352}$ AS, Strep(II)-MAP1S LC WT, Strep(II)-MAP1S LC S786/812A and Strep(II)-MAP1S LC S786/812D were expressed and purified in insect cells. Typically, 1 l of Sf9 insect cells at $2 \times 10^6$ cells/ml was infected with 2 ml of high titre virus for 48 h and centrifuged at 3,000 *g* for 15 min at 4°C. Cell pellets were resuspended in 25 mM NaHCO$_3$ and lysed with 2× lysis buffer containing 100 mM Tris–HCl pH 7.5, 1 M NaCl, 30 mM imidazole pH 8.0, 5% glycerol, 10 mM MgSO$_4$, 0.3% Triton X-100, 2 mM TCEP, 1× protease inhibitor cocktail (Roche) and 1:1,000 nuclease. Lysate was centrifuged at 55,000 *g* for 30 min at 4°C. Supernatant containing solubilized His$_6$-CDKL5$^{1–352}$ was loaded on Hispur cobalt resin (Thermo Scientific) and washed three times with wash buffer containing 50 mM Tris–HCl pH 7.5, 500 mM NaCl, 5% glycerol, 1 mM TCEP, 0.01% Triton X-100 and 15 mM imidazole. Bound proteins were eluted in 1 ml wash buffer containing 500 mM imidazole. Protein concentration was measured by comparison with BSA standards. Strep(II)-MAP1S LC WT, S786/812A and S786/812D were purified similarly to the procedure described above using a Strep-Tactin column.

### *In vitro* kinase assays

Purified proteins were incubated with kinase buffer containing 20 mM Tris–HCl pH 7.5, 10 mM MgCl$_2$, 1× protease inhibitor cocktail (Roche), 1 μM okadaic acid, 1 mM DTT, 100 μM ATP and 0.5 mM ATPγS or 6-benzyl-ATPγS (Biolog) at 30°C. Analog-sensitive CDKL5 auto-thiophosphorylation was determined with ~250 ng (200 nM) CDKL5$^{1–352}$ on HA-beads for 30 min. All substrate thiophosphorylation was done with insect cell-purified AS-CDKL5, 6-benzyl-ATPγS and 150 ng (50 nM) MAP1S, 200 ng (170 nM) EB2, 150 ng (50 nM) ARHGEF2 or 300 ng (130 nM) AMPH1. Time courses were performed with 50 ng (40 nM) AS-CDKL5 for 0–60 min. Titrations with 0–50 ng AS-CDKL5 incubated for 30 min (MAP1S, EB2) or 60 min (AMPH1, ARHGEF2). Phosphomutant experiments incubated as long as the titrations and contained analog-sensitive and kinase-dead CDKL5$^{1–352}$ at the following concentrations: 50 ng (AMPH1), 5 ng (MAP1S), 10 ng (EB2), 20 ng (ARHGEF2). Reactions were quenched with 20 mM EDTA and followed with alkylation by 5 mM p-nitrobenzyl mesylate (Abcam) for 1 h at RT. Proteins were solubilized by sample buffer containing 0.1 M DTT. Thiophosphorylation was detected by anti-thiophosphate ester antibody on Western blots.

### Substrate labelling and covalent capture

Direct CDKL5 substrates were isolated by covalently capturing thiophosphorylated peptides based on a method described before (Hertz *et al*, 2010). Full P12 mouse brain was lysed in lysis buffer containing 20 mM Tris–HCl pH7.5, 100 mM NaCl, 10 mM MgCl2, 0.5 mM DTT, 1× Protease Inhibitor Cocktail (Roche), 1 μM okadaic acid and 0.25% IGEPAL NP-40. Lysates were sonicated briefly three times and incubated for 30 min on ice. Lysates were subsequently centrifuged at 20,000 *g* for 15 min at 4°C. Total protein concentration was measured using BCA protein assay (Pierce). 2.5 mg total protein P12 mouse brain lysate was labelled by 3–5 μg AS or KD CDKL5 using a kinase labelling mix including 1 μM PKA inhibitor, 0.2 μM PKC inhibitor, 3 mM GTP, 100 μM ATP, 0.5 mM DTT, 0.5 μM okadaic acid and 0.5 mM 6-benzyl-ATPγS

for 1 h at 30°C on a nutator. Samples were denatured with 60% urea and 10 mM TCEP for 1 h at 55°C. After dilution with 2× volume 50 mM ammonium bicarbonate while keeping TCEP at 10 mM, samples were digested by 30 μg Sequencing Grade Modified Trypsin (Promega) at pH 8.0 rotating o/n at 37°C. Trypsinization was quenched by 0.5–1.0% trifluoroacetic acid. Clean-up of the digested peptides is done with Sep-Pak Classic C18 Cartridges (Waters) and completely dried on a Speedvac. Dried samples are resuspended in 50% acetonitrile and 50 mM HEPES, set to pH 7.0 and incubated with Sulfolink Coupling Resin (Thermo Scientific) and 25 μg BSA rotating o/n at RT. Resin was subsequently washed with MQ, 5 M NaCl, 50% ACN and 5% formic acid, respectively, then incubated with 10 mM DTT for 20 min, and eventually thiophosphorylated peptides were eluted with Oxone after 30 min of incubation. Final pre-MS desalting and clean-up steps were done with Zip Tips – P10, 0.6 μl C18 resin (Millipore), and purified peptides were flash frozen in liquid N2 to be analysed with mass spectrometry. Three independent experiments were performed totalling 8 AS and 6 KD replicates. Exp. 1: 2 AS, 1 KD. Exp. 2: 2 AS, 2 KD. Exp. 3: 4 AS, 3 KD.

### Mass spectrometry

Liquid chromatography-tandem mass spectrometry (LC-MS/MS) was used to analyse direct thiophosphorylation of substrates by AS-CDKL5. Each sample was resuspended in 1% TFA, sonicated for 15 min and injected 2 (Exp. 1 and 2) or 3 times (Exp. 3). Peptide mixtures were separated on a 50 cm, 75 μm I.D. Pepmap column over a 3-h gradient and eluted directly into the mass spectrometer (Orbitrap Velos). Xcalibur software was used to control the data acquisition. The instrument ran in data-dependent acquisition mode with the top 10 most abundant peptides selected for MS/MS by CID, MSA or HCD fragmentation techniques (one fragmentation technique per replicate).

Data processing was performed using the MaxQuant bioinformatics suite, and protein database searching was done by the Andromeda search engine using a Uniprot database of *Mus musculus* proteins amended with common contaminants. A protein, peptide and phosphosite estimated false discovery rate of 1% was used to generate tables with protein and phosphopeptide identifications and quantifications, which featured matching between runs for peptide identification. Phosphosite tables were subsequently uploaded into the Perseus analysis program for further statistical analysis and data visualization. The following parameters were reported: peptide posterior error probabilities for identification using a target-decoy approach. For sites found on multiphosphorylated peptides, the least modified peptide ratio was reported: phosphosite localization probabilities for site identification. Putative substrate candidates were considered when repeated at least twice in AS samples and always absent in KD controls, a PEP-score lower than 10E-05 and a localization probability higher than 0.800.

### Immunocytochemistry

Cells were fixed with cold 4% PFA, 4% sucrose in PBS for 10–15 min. For EB2 staining, neurons were fixed with methanol supplemented with 1 mM EGTA for 5 min at −20°C, followed by 5 min of

4% PFA, 4% sucrose based on Jaworski *et al* Mouse neurons were blocked and permeabilized with 0.2% Triton X-100/10% normal goat serum in PBS for 60 min at RT. Human cells were treated the same with 0.1% Triton X-100/3% BSA in PBS. In blocking buffer, the following primary antibodies were incubated o/n at 4°C: mouse anti-MAP2 (1:1,000, Sigma M9942), rat anti-EB2 (1:1,000, Abcam ab45767), rabbit anti-EB2 pS222 (1:1,000), rabbit anti-MAP1S LC (1:500, Sigma HPA050934), rabbit anti-OCT4 (1:500, Abcam ab19857) and mouse anti-TRA-1-60 (1:250, Abcam ab16288). After three 5-min washes with PBS, 1:500 secondary antibodies in PBS were incubated for 2 h at RT and washed three times again with one containing 1:2,000 DAPI stain (Thermo Scientific). Coverslips were mounted with Fluoromount-G (Southern Biotech) or Prolong Gold Antifade (Invitrogen) on glass slides.

## Neuronal treatments

Neuronal depolarization was achieved by directly adding 3 M KCl to a final concentration of 50 mM. 5 M NaCl to a final concentration of 50 mM was used as a osmolarity control. Neurons were incubated at 37°C for assay-dependent lengths of time and were lysed directly in 1× sample buffer supplemented with 0.1 mM DTT. Zero-minute controls were lysed without adding reagents. Thirty minutes of pre-incubation with 100 μM D-AP5 (Abcam) was done before KCl treatment to block NMDA receptors. NMDA and OA treatments were performed similarly by either adding 100 mM NMDA to a final concentration of 50 or 500 μM OA to a final concentration of 1 μM directly to the culture media.

## Neuronal morphology

Mice were put under surgical anaesthesia with 80–100 mg/kg ketamine (Vetalar) + 10 mg/kg xylazine (Rompun) injected intraperitoneally. While deeply anaesthetized, mice were fixed by cardiac perfusion with ~0.5 ml/mg bodyweight PBS to clear the blood, followed by ~1 ml/mg bodyweight 4% PFA until stiff. The brain was removed and soaked in 4% PFA for 24 h. Fixed brains were cut into 50- to 100-μm-thick coronal sections using Leica VT1000S vibrating blade microtome and mounted on glass slides with Fluoromount-G (Southern Biotech). Complete basal dendritic arbours of Thy1-GFP labelled neurons were imaged without immunostaining in 100-μm-thick coronal brain sections. Somatosensory layer 5 pyramidal cells were localized with help of the mouse brain atlas. Only bright neurons in the middle of the cross-section were imaged to prevent dendrites being cut. *Z*-stacks were acquired with a Leica SP5-inverted confocal microscope and a Leica HyD photodetector (20×/0.5 NA oil, 1.5× optical zoom, 1.5 μm *z*-step size, 2 airy pinhole). Images were processed with Leica Application Suite and Fiji. Basal dendritic arbours were reconstructed using Neurolucida 360 (MBF Bioscience) and analysed with Neurolucida Explorer (MBF Bioscience). Three-dimensional total dendrite length was calculated using Fiji plugin NLMorphologyViewer. For dendritic spine analysis, basal secondary dendrites of Thy1-YFP labelled neurons were imaged without immunostaining in 50-μm-thick coronal brain sections. Somatosensory layer 5 pyramidal cells were localized with help of the mouse brain atlas. *Z*-stacks were acquired with a Leica SP5-inverted confocal microscope and a Leica HyD photodetector (100×/1.46 NA oil, 2× optical zoom, 0.25 μm *z*-step

size, 1 airy pinhole). Images were processed with Leica Application Suite and Fiji. Spine density and head diameter were manually measured using a custom Fiji plugin.

Mouse primary neuronal cultures were transfected with 0.5 μg GFP at DIV7 (dendrites) or DIV11 (spines) using Lipofectamine-2000 (Invitrogen) according to manufacturer's instructions. At DIV11 (dendrites) or DIV18 (spines), neurons were fixed with 4% PFA for 15 min at RT and permeabilized with 0.3% Triton X-100 in 10% normal goat serum. Primary antibodies incubated at 4°C o/n and fluorescent secondary antibodies (Jackson Immunoresearch) incubated at RT for 2 h. Cells were counterstained with DAPI (Thermo Scientific) and mounted with Fluoromount-G (Southern Biotech). The following antibodies were used for immunocytochemistry: rat anti-Ctip2 (1:2,000, Abcam), chicken anti-GFP (1:2,000, Aves). Pyramidal neurons were identified by typical morphological hallmarks like the presence of an axon and a branched dendritic arbour. Hippocampal CA3 neurons were specifically selected by negative Ctip2 staining. For dendrite morphology, images were acquired with an Olympus IX83 widefield microscope and a Hamamatsu ORCA-Flash4.0 camera in one focal plane (20×/0.75 NA air). Images were processed in Olympus cellSens Dimension and Fiji. Dendrites were reconstructed using Neurolucida 360 (MBF Bioscience) and analysed with Neurolucida Explorer (MBF Bioscience). For dendritic spine analysis, *z*-stacks of secondary dendrites were acquired with a Leica SP5-inverted confocal microscope and a Leica HyD photodetector (63×/1.4 NA oil, 4× optical zoom, 1 μm *z*-step size, 1 airy pinhole). Images were processed with Leica Application Suite and Fiji. Spine density was manually measured using a custom Fiji plugin.

## Microtubule dynamics

Live imaging of EB3-tdTomato expressing primary neurons at DIV14 was done at 37°C and 5% $CO_2$ in 35-mm glass-bottom dishes containing dye-free media. Thirty-six hours prior to imaging, mouse primary cortical cultures were transfected with 1 μg EB3-tdTomato and 1 μg shRNA-containing or empty pLL3.7-EGFP using Lipofectamine-2000 (Invitrogen) according to manufacturer's instructions. Flat stretches of dendrite were selected based on their thickness and protein expression levels. Three-minute videos were acquired at 1 frame/s with an Olympus IX81 spinning disc confocal microscope and a Hamamatsu C9100-13 CCD camera in one focal plane (150×/1.45 NA oil). Videos were processed using Volocity and Fiji. Background correction was applied by subtracting the average intensity. Kymographs were generated using Fiji KymographClear 2.0 and manually analysed.

## TrkB trafficking

Live imaging of TrkB-RFP expressing primary neurons at DIV14 was done at 37°C and 5% $CO_2$ in 35-mm glass-bottom dishes containing dye-free media. Seventy-two hours prior to imaging, mouse primary cortical cultures were transfected with 1 μg TrkB-RFP and 1 μg pLL3.7-EGFP using Lipofectamine-2000 (Invitrogen) according to manufacturer's instructions. Flat stretches of dendrite were selected based on their thickness and protein expression levels. Two-minute videos were acquired at 3 frame/s with an Olympus IX81 spinning disc confocal microscope and a

Hamamatsu C9100-13 CCD camera in one focal plane (150×/1.45 NA oil). Videos were processed using Volocity and Fiji. Bleach correction was applied using a simple ratio correction method with the background set at 1,400. Spots were manually tracked with Fiji TrackMate and analysed with a custom Fiji plugin. Tracks were considered dynamic when the net distance exceeded 5 μm. After intrapolation and track smoothing, retrograde and anterograde runs were selected by instantaneous velocities > 0.1 μm/s and a net distance > 2 μm without changing direction. Kymographs were generated using Fiji KymographClear 2.0.

### Microtubule colocalization

0.5 μg FLAG-CDKL5 and 0.5 μg HA-MAP1S LC were co-expressed in COS7 cells for 36 h using Xtremegene 9 (Roche) according to manufacturer's instructions. Coverslips were fixed with methanol for 5 min at −20°C and permeabilized with 0.3% Triton X-100 in 10% normal goat serum. Primary antibodies incubated at 4°C o/n, and fluorescent secondary antibodies (Jackson Immunoresearch) incubated at RT for 2 h. Cells were counterstained with DAPI (Thermo Scientific) and mounted with Fluoromount-G (Southern Biotech). The following primary antibodies were used for immunocytochemistry: rat anti-FLAG (1:100, Thermo Scientific MA1-142), rabbit anti-MAP1S LC (1:500, Sigma HPA050934), mouse anti-αTubulin (1:1,000, Sigma T9026). High CDKL5-expressing cells were selected for healthy nuclei and a distinct microtubule cytoskeleton. Multichannel images were acquired with an Olympus IX83 widefield microscope and a Hamamatsu ORCA-Flash4.0 camera in one focal plane (60×/1.42 NA oil). Images were processed with Olympus cellSens Dimension and Fiji. Colocalization of MAP1S and microtubules was quantified binary (colocalization or no colocalization) manually and by Pearson coefficient measured with HK means segmentation and Colocalization Studio in Icy.

### Microtubule co-sedimentation

Five microgram purified Strep(II)-MAP1S LC WT, S786/812A or S786/812D was thiophosphorylated by 0.5 μg AS-CDKL5[1–352] with 0.5 mM benzyl-ATPγS in kinase buffer for 45 min at 30°C. kinase-dead CDKL5[1–352] was used as a negative control. Co-sedimentation of taxol-stabilized microtubules with thiophosphorylated MAP1S LC was based on a method described before (Campbell & Slep, 2011). Taxol-stabilized microtubules were prepared by incubating 20 μM purified tubulin protein (Cytoskeleton) with 1 mM GTP and DTT in BRB80 buffer, while gradually increasing the taxol concentration to 20 μM at 37°C. 10 μM of taxol-stabilized microtubules was incubated with 3 μM phosphorylated or non-phosphorylated MAP1S LC for 20 min at 25°C (3:1 ratio). 3 μM MTs with 3 μM MAP1S LC were used for the 1:1 ratio. 10% of the protein samples was saved as "input", while the rest was loaded on top of a 40% glycerol cushion in an ultracentrifuge tube and centrifuged at 100,000 g for 30 min at 25°C. The supernatant was saved as the "soluble" fraction, and the pellet as the "MT-bound" fraction after washing three times with BRB80. Proteins were solubilized by sample buffer containing 0.1 M DTT. MAP1S LC binding to microtubules and phosphorylation state was detected by anti-MAP1S LC and phosphospecific antibodies on Western blots.

### Statistical analysis

Data were analysed using GraphPad Prism 7. Exact values of $n$ and statistical methods are mentioned in the figure legends. A $P$-value lower than 0.05 was considered statistically significant. All error bars in figures are SEM.

## Data availability

The mass spectrometry proteomics data have been deposited to the ProteomeXchange Consortium via the PRIDE (Vizcaino et al, 2016) partner repository with the dataset identifier PXD010511 (https://www.ebi.ac.uk/pride/archive/projects/PXD010511).

**Expanded View** for this article is available online.

## Acknowledgements

We would like to thank David Barry and Donald Bell of the Light Microscopy Science Technology Platform at the Francis Crick Institute for their input and Richard Li for his help with design of CDKL5 second-site rescue mutations. This work was supported by the Francis Crick Institute which receives its core funding from Cancer Research UK (FC001201), the UK Medical Research Council (FC001201) and the Wellcome Trust (FC001201); LLB was the recipient of a 2017 Loulou Foundation Junior Fellowship. SKU was supported by grants from the International Foundation for CDKL5 Research (FC001003) and the Loulou Foundation (CDKL5-17-109-01). ARM was supported by grants from the International Foundation for CDKL5 research (20152375), the Loulou Foundation (3823735) and a NIMH R21MH10771.

### Author contributions

LLB, PDN, ARM and SKU designed the experiments and wrote most of the manuscript. LLB and PDN conducted most of the experiments and analyses. MS and MM conducted some of the experiments. SC contributed to multiple experiments. EC contributed to protein purification experiments. HRF and APS helped design and conduct mass spectrometry experiments.

### Conflict of interest

A.R.M. is a co-founder and has equity interest in TISMOO, a company dedicated to genetic analysis focusing on therapeutic applications customized for autism spectrum disorder and other neurological disorders with genetic origins. The terms of this arrangement have been reviewed and approved by the University of California San Diego in accordance with its conflict of interest policies.

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
