## [Review Process File · The EMBO Journal]

Chemical genetic identification of CDKL5 substrates reveals its role in neuronal microtubule dynamics

Lucas L. Baltussen, Priscilla D. Negraes, Margaux Silvestre, Suzanne Claxton, Max Moeskops, Evangelos Christodoulou, Helen R. Flynn, Ambrosius P. Snijders, Alysson R. Muotri, Sila K. Ultanir

Review timeline:

Submission date:	3rd May 2018
Preliminary Editorial Decision:	8th Jun 2018
Final Editorial Decision:	18th Jun 2018
Revision received:	20th Jul 2018
Accepted:	31st Aug 2018

Editor: Hartmut Vodermaier

Transaction Report:

Preliminary Editorial Decision

8th Jun 2018

Thank you again for submitting your manuscript on CDKL5 substrate identification to The EMBO Journal. I apologize for the delay in getting back to, owed to the fact that one of the three reviewers who agreed to evaluate the study has still not returned their report. To avoid further loss of time, I have therefore decided to send you at least a preliminary decision based on the two reports at hand, which I am forwarding to you copied below. As you will see, reviewers 1 and 3 are both generally appreciative of the work, but referee 3 nevertheless brings up a number of concerns regarding the quality/conclusiveness of some of the data. Should you be able to decisively clarify these issues to the referees' satisfaction, we should be happy to consider a revised version of the manuscript further. Since it is our policy to allow only a single round of major revision, it will however be important to carefully respond to all points of the two reviewers at this stage, and I should also stress that this preliminary decision may still remain subject to change should the outstanding third report come in over the next days and raise substantive additional concerns. In any case, please be assured that our final assessment of your study will not be affected by publication of related work here or elsewhere, and that we may therefore also discuss a possible extension of the revision deadline in case this should be helpful.

REFeree REPORTS:

Referee #1:

The very well written study by Baltussen and colleagues uses a number of convincing approaches to analyse and validate substrates of CDKL5 that are involved in neuronal microtubule function and dynamics. They first discover, using a chemical genetic approach with a doubly mutated catalytically 'rescued' CDKL5 and mouse brain lysate that CDKL5 phosphorylates several MT-associated proteins, and that MAP1S is likely a physiological phosphorylated substrate, with

phosphorylation probably controlling association with MTs (based on data with mutants, phosphospecific antibodies and microscopic and biochemical analysis). When combined with CDKL5 murine knockout and human patient data (where EB2 phosphorylation is markedly, and significantly, decreased), this forms a compelling and very interesting story ideal for publication in EMBO J. The finding that an RPXSA motif is possibly predictive of CDKL5 phosphorylation also really pushes the CDKL5 field forward, providing an opportunity to get to grips with several disease (are there more CDKL5 substrates waiting to be uncovered?) and to generate new potential biomarkers for analysing CDKL5 small molecule modulators. More of this could be made in the discussion.

Major points.

- 1) In Figure 5, why was the tyrosine kinase Trkb]B (fused to RFP) the cargo chosen for monitoring in terms of transport? Would any other trafficked substrate have behaved similarly in terms of run-length? Could an endogenous protein be used instead?
- 2) Figure 6E shows a MT co-localisation assay. Given the small % of MAP1S OC that binds to tubulin, the output data would be more convincing if binding were rescued by a S786D/S812D (or EE) mutant. Has this experiment been attempted? At the minimum, is the MAP1S in the supernatant actually phosphorylated at S812, based on using the pS182 phosphospecific antibody (Figure 2)

Minor points.

- 1) Figure 1 and Figure EV1 are excellent, but for non-experts, panels E and F in EV1 and D in Fig 1 are only very quickly mentioned in the legend. It would be useful to accurately explain (in the body of the paper) why the b and y ion series detected confirm phosphorylation at the second S residue (rather than the S or T) in the annotated spectra for ARHGEF2 and EB2, which allows the RPTS(p)A motif to be identified.
- 2) It will naturally be appropriate for this manuscript to cite related work identifying MAP1S (Munoz et al, submitted to EMBO J), should it be published before or at the same time as this study.
- 3) The tense shifts from present to past tense throughout the manuscript, which should be corrected.

Referee #3:

Review comments:

This manuscript submitted by Baltussen et al. described new CDKL5 substrates and their role in neuronal microtubule dynamics. The authors applied a chemical genetics approach, using ATP analog benzyl-ATP γ S which is bulkier than usual ATP and a mutated, analog-specific CDKL5, to identify substrates phosphorylated during the incubation of purified CDKL5 with whole cell lysate. Benzyl-ATP γ S can be incorporated into potential substrates, and subsequently digested, captured by thio-reactive beads, and eluted with oxidation. Substrate phosphorylation sites were identified by LC-MS/MS. The authors generated antibodies recognizing identified substrate phosphosites from EB2 and MAP1S. These antibodies were used to localize and quantify *in vivo* phosphorylation in mice cortex by imaging and western blots. Further relationships were established between neuronal activity, microtubule dynamics, cargo trafficking and CDKL5 as well as its substrate proteins EB2 and MAP1S, through phosphorylation regulation. While the above-mentioned studies were done in mouse, the authors validated phosphorylation sites in iPS-derived human neurons. The amount of work presented in this manuscript is certainly not trivial, however, I'm concerned about the quality of data used for supporting these conclusions. I expect more solid evidence to warrant publication.

1. Fig 1D and Fig EV1F MS/MS spectra look poor, as many fragment peaks were picked from noise and a number of abundance peaks were not assigned. Some peaks in Fig 1D might be neutral loss of phospho group, they need to be annotated. Also need to show precursor mass error for correct assessment of PSM matching quality.

From my interpretation based on fragment type, Fig EV1-E, F should be all CID spectra from Velos. Based on iontrap CID 1/3 rule, fragment peaks below 1/3 of precursor m/z should be suppressed and low in abundance. However, these two spectra all have abundant peaks below 200 Da. Please

explain why they present.

2. Extracted ion chromatogram from mass spec needs to be shown for the three substrate phosphopeptides for all replicates, between control and AS-CDKL5 treated samples. This is a more convincing data to demonstrate presence or absence of these phosphorylation sites under different conditions.
3. Figure 1A is not clear. If the negative control is kinase dead (KD) CDKL5, it should be used in the left part, rather than the 'WT kinase' in the current version, unless I misunderstand the purpose of this figure.
4. What amount of AS-CDKL5 was used in Fig 1E? What time point was used in Fig 1G?
5. Was MAP1S S812 detected? Based on Fig 1K, S812 seems to be a more prevalent phosphorylation site than S786. Could you develop a targeted MS assay to quantify this site?
6. Interesting correlation between increasing CDKL5 level and decreasing EB2 pS222 level as shown in Fig 2C. Can the authors provide additional investigation on how this is regulated?
7. Critical data are missing from the evaluation of in-house generated EB2 pS222, MAP1S pS786, pS812 pAb in terms of the specificity and cross-reactivity to other proteins. This could raise question to Fig 2G, which seems like EB2 pS222 is present everywhere in cell (or abundant in nuclei), but EB2 is excluded from nuclei. This could either be: a) nuclei fraction EB2 is a small fraction in total protein, but highly phosphorylated, or b) the phospho antibody cross reacts with other proteins.
8. Fig 2C the MAP1S pS812 band is so low that hardly can make conclusion. Fig 2D is not clear whether it's quantifying CDKL5 WT or KO. Two genotypes but only one set of plot is shown.
9. Why MAP1S shRNA knock down has different conclusion in Fig 4F (n.s.) vs. in Fig EV 5E (**)? Why different statistics was used in Fig 4F (Tukey's multiple comparison) vs. in Fig EV 5E (Dunnet's multiple comparison)? More independent shRNA nucleotides may be needed to knock down the same target protein and show consistent result.

Final 1st Editorial Decision

18th Jun 2018

We have now finally received the outstanding third report on your manuscript, which I am forwarding to you copied below. As you will see, this reviewer 2 is also overall supportive of publication - thus confirming our original decision to invite a revision here - but nevertheless raises a number of specific issues that I would ask you to address together with those of referees 1 and 3 when preparing your revised manuscript. Should you have any questions or comments in this regard, please do not hesitate to get back to me.

REFEREE REPORT:

Referee 2

Here the authors have characterized the understudied CDKL5 kinase, encoded by the CDKL5 gene on the X chromosome, in which mutations result in an early onset neurodegenerative disease defined as CDKL5 disease. To do this they exploited the chemical biology approach that they used previously to identify substrates for two other neuronal protein kinases, which involves making a recombinant form of an analogue sensitive mutant of the kinase of interest, and then using an N6-modified γ -thioATP analogue to phosphorylate proteins in a lysate of a target cell/tissue. In this case, they created an F89A C152A CDKL5 mutant and used N6 benzyl γ -thioATP to phosphorylate a lysate from P12 mouse brain, recovering thio-phosphorylated tryptic peptides after modification of the thiophosphate residues with p-nitro-benzyl-mesylate, followed by MS analysis. This led them to identify sites in three microtubule (MT) binding proteins, S786 and S812 in MAP1S, S222 in EB2, and S122 in ARHGEF2, as CDKL5 substrates. They went on to validate these sites as being directly phosphorylated by CDKL5 by making Ser to Ala mutations and by generating phosphospecific antibodies for the four sites. In transfected HEK293 cells, they found that S786 MAP1S and S122 ARHGEF2 phosphorylation was increased by over-expression of CDKL5, whereas EB2 S222 and MAP1S S812 were highly basally phosphorylated. They showed that pS812 MAP1S and EB2 pS222 were high in WT but not Cdk15^{-/-} P20 brain and that phosphorylation of these sites declined after P20, even though CDKL5 protein levels remained high in adulthood. EB2 pS222 was primarily localized to excitatory neuron dendrites, and its level was reduced by activation of the NMDA

receptor. Their identification of MT-binding proteins as CDKL5 substrates led them to test whether CDKL5 activity regulates MT function. They found a small reduction in total dendrite length in *Cdkl5*^{-/-} cortical neurons, and based on live cell imaging with tdTomato-EB3 showed MT dynamics were reduced in *Cdkl5*^{-/-} neurons, and that dendritic microtubules had longer EB3-labeled plus-end growth duration in *Cdkl5*^{-/-} mouse neurons than in WT neurons, which was reversed by shRNA reduction of MAP1S levels. They also found that endosomes with TrkB-RFP cargo exhibited a reduced anterograde run length in *Cdkl5*^{-/-} neuronal dendrites. They went on to show that the strong MT localization of WT MAP1S was lost in COS-7 cells expressing WT but not kinase-dead CDKL5, and that the MT localization of S786/812A mutant MAP1S was unaffected by CDKL5 expression. This led them to conclude that CDKL5-mediated phosphorylation of MAP1S causes dissociation of MAP1S, and they confirmed this by demonstrating that *in vitro* phosphorylation of MAP1S with CDK5L reduced its binding to MTs. Finally, they showed that phosphorylation of EB2 S222 was reduced in CDKL5 mutant neurons differentiated from iPSCs derived from the cells of three human CDKL5 disease patients. They conclude that defective MT regulation in neuronal dendrites as a result of lack of CDKL5 phosphorylation of key MT regulators underlies the neural phenotypes of CDKL5 disease.

The identification of authentic substrates for the CDKL5 protein kinase would be an important step forward in providing a mechanistic understanding of why CDKL5 mutants lead to neurodevelopmental disease. The approach of using an analogue-sensitive CDKL5 mutant, combined with thiophosphorylation of proteins in tissue lysates with an N6-modified γ -thioATP, has worked for other protein kinases, and the authors' evidence that this can be used to identify authentic CDKL5 substrates is convincing. Their finding that CDKL5 proteins include several microtubule-associated substrates involved in MT function is exciting, and their demonstration that lack of CDKL5 phosphorylation in CDKL5 knockout cells results in altered MT dynamics and defects in anterograde vesicle transport points to the importance of CDKL5 function in MT regulation in neurons, and suggests that a MT defect is a cause of CDKL5 disease. The use of patient iPSC-derived neurons to establish the importance of CDKL5 in phosphorylation of the EB S222 sites is nice, although it would be even better if they had used these human cells to re-express WT or kinase-dead CDL5 and investigate the defects in MT trafficking and neuronal dendritic function. One other weakness is that there is no investigation of the possible functional consequences of CDKL5 phosphorylation of ARHGEF2 S122 or EB2 S222, although there are already a lot of data in the paper.

Points: 1. Figure 1M: The primary consensus motif for CDKL5 is rather minimal - how many RPXS sites are there in the proteome. Based on the four identified sites, it appears that CDKL5 is not a proline-directed kinase, even though CDKL5 lies in the CMGC branch of the kinome, where the majority of kinases are Pro-directed. Is there any evidence for the primary sequence specificity of other kinases on the CDKL branch, and does CDKL5 require an accessory subunit *in vivo*?

2. Given that the primary CDKL5 consensus motif is quite minimal, one wonders whether there is any crossreaction of the site-specific phosphoantibodies the authors generated between the different sites, e.g. between pS122 ARHGEF2 and pS222 EB2. In this regard, it would be important for the whole MW range to be shown for the blots with the different phosphoantibodies, in supplementary figures.

3. Page 5: Based solely on the observed decrease in EB2 pS222 and MAP1S pS812 levels, the authors are not really justified in concluding that there is a decrease in CDKL5 activity beyond P20. There are certainly other explanations. For instance, the level of protein phosphatase activity towards these phosphosites might increase significantly beyond P20. Alternatively, expression of a CDKL5 activator could decline beyond P20.

4. Figure 3D: The CDKL5 bands should be quantified in this panel as well,

5. Figure 4: Re-expression of MAP1S S786/812A or EB2 S222A in the MAP1S or EB2 knockdown neurons would provide direct evidence that CDKL5 phosphorylation of these sites is important in the observed phenotypes.

6. Figure 6E: The fraction of MAP1S that failed to bind to MTs as a result of CDKL5 phosphorylation *in vitro* was apparently rather small - did the authors estimate the phosphorylation

stoichiometry of the S786 and S812 sites. If it was very low, this could explain the relatively small effect. Did the authors consider testing a MAP1S S786/812E or D mutant for in vitro MT binding studies, or in vivo?

7. Figure 7: It would help the reader if the authors described in the text the types of CDKL5-deficiency mutations present in these three patients.

8. As a way to rapidly turn CDKL5 function on and off in target cells, did the authors consider expressing the AS-CDKL5 mutant in CDKL5-null cells and then use a suitable AS inhibitor to turn its activity on and off to examine effects on MT function?

1st Revision - authors' response

20th Jul 18

Referee #1

The very well written study by Baltussen and colleagues uses a number of convincing approaches to analyse and validate substrates of CDKL5 that are involved in neuronal microtubule function and dynamics. They first discover, using a chemical genetic approach with a doubly mutated catalytically 'rescued' CDKL5 and mouse brain lysate that CDKL5 phosphorylates several MT-associated proteins, and that MAP1S is likely a physiological phosphorylated substrate, with phosphorylation probably controlling association with MTs (based on data with mutants, phosphospecific antibodies and microscopic and biochemical analysis). When combined with CDKL5 murine knockout and human patient data (where EB2 phosphorylation is markedly, and significantly, decreased), this forms a compelling and very interesting story ideal for publication in EMBO J. The finding that an RPXSA motif is possibly predictive of CDKL5 phosphorylation also really pushes the CDKL5 field forward, providing an opportunity to get to grips with several disease (are there more CDKL5 substrates waiting to be uncovered?) and to generate new potential biomarkers for analysing CDKL5 small molecule modulators. More of this could be made in the discussion.

Major points.

1) In Figure 5, why was the tyrosine kinase TrkB (fused to RFP) the cargo chosen for monitoring in terms of transport? Would any other trafficked substrate have behaved similarly in terms of run-length? Could an endogenous protein be used instead?

We thank the referee for this comment. We have chosen TrkB-RFP as a model cargo for investigation of microtubule based trafficking. This is mainly because TrkB trafficking in axons and dendrites have been studied by multiple groups over many years. This allowed us to compare our data to previous results and establish that our data (e.g. speed, % retrograde movement) is comparable to previous reports. We have now indicated this observation in Results section of TrkB transport. In addition, TrkB trafficking in dendrites rely on both kinesin and dynein motors for transport, not limiting our observations to a specific motor type and making it a good model cargo for our study. This reasoning is now clarified at the first paragraph of the Results section on TrkB. Other cargo may be affected differently depending on their usage of motor proteins. Our study provides an initial result showing that some cargo transport is affected in CDKL5 KO mice. Imaging trafficking of endogenous TrkB trafficking is not currently feasible to the best of our knowledge. We believe that a more comprehensive investigation of multiple different cargo proteins is very important, however due to time limitations in obtaining these data, beyond the scope of this paper.

A sentence is now added to the Discussion section "CDKL5 as a regulator of microtubule dynamics and trafficking": "Different cargo proteins may be affected differently in CDKL5 KO animals and more experiments are needed to determine the endogenous cargo proteins that may be affected."

2) Figure 6E shows a MT co-localisation assay. Given the small % of MAP1S OC that binds to tubulin, the output data would be more convincing if binding were rescued by a S786D/S812D (or EE) mutant. Has this experiment been attempted? At the minimum, is the MAP1S in the supernatant actually phosphorylated at S812, based on using the pS182 phosphospecific antibody (Figure 2)?

We have now generated a phosphomimetic MAP1S LC (S786/812D). We show that the putative phosphomimetic behaves similar to MAP1S phosphomutant (S786/812A). 1) in COS7 cells overexpression of HA tagged MAP1S LC S786/812D leads to its localization on microtubules (now

shown in appendix Figure S5D), 2) in microtubule pelleting assay MAP1S S786/812D binds and precipitates with microtubules similar to MAP1S LC S786/812A (now shown in appendix Figure S5C). These indicate that, in this case S786/812D phosphomimetic does not act like a true phosphomimetic protein. The failure of phosphomimetics is observed frequently as Aspartic Acid or Glutamic Acid are not perfect substitutes for a phosphorylated Serine/Threonine.

The % of MAP1S LC that does not bind tubulin is small (referee made a typo here). The % of MAP1S LC that binds to tubulin depends on the amount of polymerized tubulin that is used and other conditions of our in vitro assay e.g. temperature etc. We now added a lower amount of tubulin to have a lower tubulin : MAP1S LC ratio. This resulted in a highly increased % unbound / bound MAP1SLC (now shown in appendix Figure S5C). We also show that the phosphorylation of both MAP1S sites S786 and S812 is present in all fractions (supernatant and pellet) (now shown in appendix Figure S5A&B). This is likely due to saturation of MAP1S phosphorylation by CDKL5 in vitro. Indeed, we have shown that phosphorylation of MAP1S with CDKL5 saturates before 30 min in our assay conditions (Figure 1E). We have used comparable in vitro kinase assay conditions for phosphorylating MAP1S WT by CDKL5 in 30 min in order to achieve a maximal phosphorylation. Therefore, we believe that all MAP1S is essentially phosphorylated in the microtubule binding assay experiment. Our results clearly demonstrate that phosphorylation of MAP1S reduces its ability to bind microtubules in vitro (Figure 6E) and that upon altering the in vitro microtubule pelleting conditions more or less binding can be obtained for phosphorylated form. This new data is also explained in the Results section "CDKL5's phosphorylation of MAP1S light chain inhibits its microtubule binding".

Minor points.

1) Figure 1 and Figure EV1 are excellent, but for non-experts, panels E and F in EV1 and D in Fig 1 are only very quickly mentioned in the legend. It would be useful to accurately explain (in the body of the paper) why the b and y ion series detected confirm phosphorylation at the second S residue (rather than the S or T) in the annotated spectra for ARHGEF2 and EB2, which allows the RPTS(p)A motif to be identified.

We agree with the referee that the mass spectrometry in figure 1 and EV1 was underrepresented. In the revised manuscript we therefore gave more attention to this data in Figure EV1 and Appendix Figure S2 and S3. The legend of Figure EV1 now contains the following explanation: "Sequences are determined by product ions of b- and y-series after collision-induced dissociation (CID) or higher-energy C-trap dissociation (HCD). Neutral loss of phosphoric acid generates a product ion series that corresponds only to those ions containing the phosphorylated residue (*)." However, we also specified here that these spectra are based on the best identification of the peptide. Therefore, the best localisation of the phosphate is not necessarily obtained from the presented spectrum.

2) It will naturally be appropriate for this manuscript to cite related work identifying MAP1S (Munoz et al, submitted to EMBO J), should it be published before or at the same time as this study. We are happy to cite and comment on unpublished related work, but we have not seen the manuscript Munoz et al. We therefore ask the editor's advice.

3) The tense shifts from present to past tense throughout the manuscript, which should be corrected. We thank the referee for pointing this out and have corrected the tense shifts throughout the revised manuscript to the best of our knowledge.

Referee #2

Here the authors have characterized the understudied CDKL5 kinase, encoded by the CDKL5 gene on the X chromosome, in which mutations result in an early onset neurodegenerative disease defined as CDKL5 disease. To do this they exploited the chemical biology approach that they used previously to identify substrates for two other neuronal protein kinases, which involves making a recombinant form of an analogue sensitive mutant of the kinase of interest, and then using an N6-modified γ -thioATP analogue to phosphorylate proteins in a lysate of a target cell/tissue. In this case, they created an F89A C152A CDKL5 mutant and used N6 benzyl γ -thioATP to phosphorylate a lysate from P12 mouse brain, recovering thio-phosphorylated tryptic peptides after modification of the thiophosphate residues with p-nitro-benzyl-mesylate, followed by MS analysis. This led them to identify sites in three microtubule (MT) binding proteins, S786 and S812 in MAP1S, S222 in EB2, and S122 in ARHGEF2, as CDKL5 substrates. They went on to validate these sites as being directly phosphorylated by CDKL5 by making Ser to Ala mutations and by generating phosphospecific antibodies for the four sites. In transfected HEK293 cells, they found that S786 MAP1S and S122

ARHGEF2 phosphorylation was increased by over-expression of CDKL5, whereas EB2 S222 and MAP1S S812 were highly basally phosphorylated. They showed that pS812 MAP1S and EB2 pS222 were high in WT but not Cdkl5^{-/-} P20 brain and that phosphorylation of these sites declined after P20, even though CDKL5 protein levels remained high in adulthood. EB2 pS222 was primarily localized to excitatory neuron dendrites, and its level was reduced by activation of the NMDA receptor. Their identification of MT-binding proteins as CDKL5 substrates led them to test whether CDKL5 activity regulates MT function. They found a small reduction in total dendrite length in Cdkl5^{-/-} cortical neurons, and based on live cell imaging with tdTomato-EB3 showed MT dynamics were reduced in Cdkl5^{-/-} neurons, and that dendritic microtubules had longer EB3-labeled plus-end growth duration in Cdkl5^{-/-} mouse neurons than in WT neurons, which was reversed by shRNA reduction of MAP1S levels. They also found that endosomes with TrkB-RFP cargo exhibited a reduced anterograde run length in Cdkl5^{-/-} neuronal dendrites. They went on to show that the strong MT localization of WT MAP1S was lost in COS-7 cells expressing WT but not kinase-dead CDKL5, and that the MT localization of S786/812A mutant MAP1S was unaffected by CDKL5 expression. This led them to conclude that CDKL5-mediated phosphorylation of MAP1S causes dissociation of MAP1S, and they confirmed this by demonstrating that in vitro phosphorylation of MAP1S with CDK5L reduced its binding to MTs. Finally, they showed that phosphorylation of EB2 S222 was reduced in CDKL5 mutant neurons differentiated from iPSCs derived from the cells of three human "CDKL5 disease patients. They conclude that defective MT regulation in neuronal dendrites as a result of lack of CDKL5 phosphorylation of key MT regulators underlies the neural phenotypes of CDKL5 disease.

The identification of authentic substrates for the CDKL5 protein kinase would be an important step forward in providing a mechanistic understanding of why CDKL5 mutants lead to neurodevelopmental disease. The approach of using an analogue-sensitive CDKL5 mutant, combined with thiophosphorylation of proteins in tissue lysates with an N6-modified γ -thioATP, has worked for other protein kinases, and the authors' evidence that this can be used to identify authentic CDKL5 substrates is convincing. Their finding that CDKL5 proteins include several microtubule-associated substrates involved in MT function is exciting, and their demonstration that lack of CDKL5 phosphorylation in CDKL5 knockout cells results in altered MT dynamics and defects in anterograde vesicle transport points to the importance of CDKL5 function in MT regulation in neurons, and suggests that a MT defect is a cause of CDKL5 disease. The use of patient iPSC-derived neurons to establish the importance of CDKL5 in phosphorylation of the EB S222 sites is nice, although it would be even better if they had used these human cells to re-express WT or kinase-dead CDKL5 and investigate the defects in MT trafficking and neuronal dendritic function. One other weakness is that there is no investigation of the possible functional consequences of CDKL5 phosphorylation of ARHGEF2 S122 or EB2 S222, although there are already a lot of data in the paper.

1. Figure 1M: The primary consensus motif for CDKL5 is rather minimal - how many RPXS sites are there in the proteome. Based on the four identified sites, it appears that CDKL5 is not a proline-directed kinase, even though CDKL5 lies in the CMGC branch of the kinome, where the majority of kinases are Pro-directed. Is there any evidence for the primary sequence specificity of other kinases on the CDKL branch, and does CDKL5 require an accessory subunit in vivo?

There are 4976 RPXS sites in human proteome and 1011 of these are known to be phosphorylated in humans (Phosphosite.org). We have now included these numbers in the first Discussion section saying "Phosphorylated RPXS motif is present in more than 1000 reported phosphorylation sites (phosphosite.org). Therefore, many more CDKL5 substrates could be present and these remain to be discovered and validated."*

There is no evidence for other CDKL kinases' specificity and no accessory subunit is identified so far. Our data shows that, in vitro, CDKL5 phosphorylates this consensus without any accessory proteins. Binding partners of CDKL5, particularly of C-terminus is an interesting open question. ICK is a kinase in CMGC family, closely related to CDKLs. ICK phosphorylation consensus sequence is RPX(S/T)(P/A/T/S) (Wu et al JBC 2012). Therefore, CDKL branch differs from the proline-directed MAPKs and CDK family of kinases and they are not necessarily Proline-directed.

2. Given that the primary CDKL5 consensus motif is quite minimal, one wonders whether there is any crossreaction of the site-specific phosphoantibodies the authors generated between the different sites, e.g. between pS122 ARHGEF2 and pS222 EB2. In this regard, it would be important for the

whole MW range to be shown for the blots with the different phosphoantibodies, in supplementary figures.

We thank the reviewer for their comment. Whole MW range blots of mouse brain lysates probed with EB2, phospho EB, MAP1S LC and phospho MAP1S are now added to Figure EV2 D&E, as stated before. As seen in these blots, phospho-specific antibodies do recognize other bands, however only the correct MW band (matching the size detected using total antibodies) shows any reduction in CDKL5 KO brains. We did not detect any specific signal with ARHGEF2 phosphospecific antibody in brain lysates. In addition, although the consensus sequence is quite minimal, phosphospecific antibodies are raised against specific 13 amino acid peptides that are not conserved between the CDKL5 substrates. Cross-reaction between the substrates is therefore unlikely.

3. Page 5: Based solely on the observed decrease in EB2 pS222 and MAP1S pS812 levels, the authors are not really justified in concluding that there is a decrease in CDKL5 activity beyond P20. There are certainly other explanations. For instance, the level of protein phosphatase activity towards these phosphosites might increase significantly beyond P20. Alternatively, expression of a CDKL5 activator could decline beyond P20.

We thank the reviewer for this comment. As also stated in response to referee 3 point 6: in this revised manuscript, we tested a second hypothesis with new experiments. We reasoned that increased phosphatase activity could result in reduced CDKL5 substrate phosphorylations in later development. "We treated cultured neurons at younger (DIV11) or older (DIV22) developmental stages with PP1/PP2A type phosphatase inhibitor, Okadaic acid (OA). We found that O.A. increases EB2 pS222 levels in both stages. The percent increase in EB2 pS222 was higher in DIV22 when compared to DIV11, indicating that PP1/PP2A activity may be higher in older stages than in younger neurons. However, even in the presence of OA the observed decrease in EB2 phosphorylation in older neurons remained, indicating that additional factors may contribute to observed reduction in pS222 (data now included in EV 3A & B)."

We agree that concluding "a decrease in CDKL5 activity in later ages" is not justified. We have now rephrased it and clarified that there may be multiple explanations for the reduced CDKL5 substrate phosphorylations, including NMDAR signalling, phosphatase activity and developmental regulation co-activators. These points are now included in Discussion stating: "NMDAR signalling, phosphatase activity or developmental regulation of putative co-activators are among possible factors that reduce these phosphorylations in adult."

4. Figure 3D: The CDKL5 bands should be quantified in this panel as well.

We agree with the referee that the Western blot in Figure 3D should have been quantified. In the revised manuscript we have therefore expanded this experiment towards n = 3 and replaced the original blot with a more representative one. We have quantified both EB2 phosphorylation and CDKL5 levels for these blots and added this information as Figure 3E.

5. Figure 4: Re-expression of MAP1S S786/812A or EB2 S222A in the MAP1S or EB2 knockdown neurons would provide direct evidence that CDKL5 phosphorylation of these sites is important in the observed phenotypes.

Overexpression of microtubule binding proteins may lead to non-specific phenotypes. In fact, MAP1S overexpression is known to be toxic in neurons (Ding et al. HMG 2006). Also, in Figure 6G overexpression of phosphomutant MAP1S LC at high levels lead to highly stabilized microtubules, and completely abnormal dendritic microtubules. We currently do not have tools to express constructs, such as MAP1S, at endogenous levels. We plan to address this comment in a future study using phosphomutant transgenic mice.

6. Figure 6E: The fraction of MAP1S that failed to bind to MTs as a result of CDKL5 phosphorylation in vitro was apparently rather small - did the authors estimate the phosphorylation stoichiometry of the S786 and S812 sites. If it was very low, this could explain the relatively small effect. Did the authors consider testing a MAP1S S786/812E or D mutant for in vitro MT binding studies, or in vivo?

As also stated in referee 1 point 2: We have now generated a phosphomimetic MAP1S LC (S786/812D). We show that the putative phosphomimetic behaves similar to MAP1S phosphomutant (S786/812A). 1) in COS7 cells overexpression of HA tagged MAP1S LC S786/812D leads to its localization on microtubules (now shown in appendix Figure S5D), 2) in microtubule pelleting assay MAP1S S786/812D binds and precipitates with microtubules similar to MAP1S LC S786/812A (now shown in appendix Figure S5C). These indicate that, in this case S786/812D

phosphomimetic does not act like a true phosphomimetic protein. The failure of phosphomimetics is observed frequently as Aspartic Acid or Glutamic Acid are not perfect substitutes for a phosphorylated Serine/Threonine.

The % of MAP1S LC that binds to tubulin depends on the amount of polymerized tubulin that is used and other conditions of our in vitro assay e.g. temperature etc. We now added a lower amount of tubulin to have a lower tubulin : MAP1S LC ratio. This resulted in a highly increased % unbound / bound MAP1SLC (now shown in appendix Figure S5C). We also show that the phosphorylation of both MAP1S sites S786 and S812 is present in all fractions (supernatant and pellet) (now shown in appendix Figure S5A&B). This is likely due to saturation of MAP1S phosphorylation by CDKL5 in vitro. Indeed, we have shown that phosphorylation of MAP1S with CDKL5 saturates before 30 min in our assay conditions (Figure 1E). We have used comparable in vitro kinase assay conditions for phosphorylating MAP1S WT by CDKL5 in 30 min in order to achieve a maximal phosphorylation. Therefore, we believe that all MAP1S is essentially phosphorylated in the microtubule binding assay experiment. Our results clearly demonstrate that phosphorylation of MAP1S reduces its ability to bind microtubules in vitro (Figure 6E) and that upon altering the in vitro microtubule pelleting conditions more or less binding can be obtained. This new data is also explained in the Results section "CDKL5's phosphorylation of MAP1S light chain inhibits its microtubule binding".

7. Figure 7: It would help the reader if the authors described in the text the types of CDKL5-deficiency mutations present in these three patients.

We thank the reviewer for this comment. We have now added the following text in Results section: "... we collected fibroblasts from three patients: p.R59X, female; p.R59X, male; D135_F154del, male and three controls. The R59X mutation leads to a premature stop codon and loss of CDKL5 protein by nonsense mediated decay, while D135_F154del leads to a truncated protein missing exon 7."

8. As a way to rapidly turn CDKL5 function on and off in target cells, did the authors consider expressing the AS-CDKL5 mutant in CDKL5-null cells and then use a suitable AS inhibitor to turn its activity on and off to examine effects on MT function?

This is a good idea and we thank the reviewer for raising this point. However, we did not have enough time for optimization of AS-kinase and AS inhibitor for use in neuronal cultures for this manuscript. Importantly, our rescued AS-CDKL5 has lost substantial amount of activity towards regular ATP γ S (Figure 1B). Therefore, our designed AS-CDKL5 would have very low levels of activity in cells to begin with. Additional mutations in ATP binding pocket would need to be explored to generate a more suitable AS-CDKL5 for AS inhibitors.

Referee #3

This manuscript submitted by Baltussen et al. described new CDKL5 substrates and their role in neuronal microtubule dynamics. The authors applied a chemical genetics approach, using ATP analog benzyl-ATP γ S which is bulkier than usual ATP and a mutated, analog-specific CDKL5, to identify substrates phosphorylated during the incubation of purified CDKL5 with whole cell lysate. Benzyl-ATP γ S can be incorporated into potential substrates, and subsequently digested, captured by thio-reactive beads, and eluted with oxidation. Substrate phosphorylation sites were identified by LC-MS/MS. The authors generated antibodies recognizing identified substrate phosphosites from EB2 and MAP1S. These antibodies were used to localize and quantify in vivo phosphorylation in mice cortex by imaging and western blots. Further relationships were established between neuronal activity, microtubule dynamics, cargo trafficking and CDKL5 as well as its substrate proteins EB2 and MAP1S, through phosphorylation regulation. While the above-mentioned studies were done in mouse, the authors validated phosphorylation sites in iPS-derived human neurons. The amount of work presented in this manuscript is certainly not trivial, however, I'm concerned about the quality of data used for supporting these conclusions. I expect more solid evidence to warrant publication.

1. Fig 1D and Fig EV1F MS/MS spectra look poor, as many fragment peaks were picked from noise and a number of abundance peaks were not assigned. Some peaks in Fig 1D might be neutral loss of phospho group, they need to be annotated. Also need to show precursor mass error for correct assessment of PSM matching quality.

From my interpretation based on fragment type, Fig EV1-E, F should be all CID spectra from Velos. Based on iontrap CID 1/3 rule, fragment peaks below 1/3 of precursor m/z should be suppressed and low in abundance. However, these two spectra all have abundant peaks below 200 Da. Please explain why they present.

We agree with the referee that not all MS/MS spectra look great. However, we believe this is a result of the stringent isolation of directly thiophosphorylated peptides by covalent capture. Later in the manuscript, we prove that the identified phosphosites are bona fide CDKL5 targets by extensive validation. Still, we improved Figure 1D and EV1E, F by combining them into Figure EV1 in the revised manuscript where spectra are better annotated and more peaks are assigned. Precursor mass errors are also now shown in the ion chromatograms of Supplementary Figure 3 (discussed in point 2).

We have made it clearer in Figure EV1 with which fragmentation method the presented spectra were obtained. The presented MAP1S spectrum is indeed fragmented by CID and analysis performed in the ion trap. However, ARHGEF2 and EB2 spectra are both fragmented by HCD and analysed with Orbitrap as the mass analyser, so the 1/3 rule does not apply to these.

2. Extracted ion chromatogram from mass spec needs to be shown for the three substrate phosphopeptides for all replicates, between control and AS-CDKL5 treated samples. This is a more convincing data to demonstrate presence or absence of these phosphorylation sites under different conditions.

We agree that extracted ion chromatograms are a convincing way to demonstrate our data. However, due to the large nature of our data set we do not think it is preferable to show chromatograms for all peptides in all replicates (multiple peptides x multiple charges x multiple injections = 117 chromatograms). Instead we chose to depict our ion chromatograms by showing the integrated peak areas for every peptide in every replicate averaged for the multiple injections. This information has been added as Appendix Figure S2. In addition, we have included the extracted ion chromatograms of one of the experiments as Supplementary Figure S3. This experiment consisted of two AS and one KD replicate, and all three peptides were identified. We therefore consider it a good example set that we hope will satisfy the reviewer's concerns. In addition, we deposited the whole dataset to the ProteomeXchange Consortium via the PRIDE partner repository with the dataset identifier PXD010511.

3. Figure 1A is not clear. If the negative control is kinase dead (KD) CDKL5, it should be used in the left part, rather than the 'WT kinase' in the current version, unless I misunderstand the purpose of this figure.

We apologise for the misunderstanding. We have altered Figure 1A to clarify that we added CDKL5 AS or KD negative control to WT mouse brain lysate. The 'WT kinase' was supposed to depict the endogenous kinases present in the brain lysate. This has now been changed to 'endogenous kinases' to prevent confusion.

4. What amount of AS-CDKL5 was used in Fig 1E? What time point was used in Fig 1G?

This information has been added to the legend of Figure 1E-H: "In vitro kinase assays showing efficient MAP1S phosphorylation by CDKL5. 50 ng (40 nM) AS-CDKL5 phosphorylates 150 ng (50 nM) MAP1S very rapidly (E, F). In 30 minutes of incubation, 150 ng MAP1S is phosphorylated by small amounts of CDKL5 (G, H)." We also added similar information for ARHGEF2 to the legend of Appendix Figure S4 A-D: "In vitro kinase assays showing efficient ARHGEF2 phosphorylation by CDKL5. 50 ng (40 nM) AS-CDKL5 phosphorylates 150 ng (50 nM) ARHGEF2 (A, B). In 60 minutes of incubation, 150 ng ARHGEF2 is phosphorylated by small amounts of CDKL5 (C, D)." And for EB2 to the legend of Appendix Figure S4 E-H: "In vitro kinase assays showing efficient EB2 phosphorylation by CDKL5. 50 ng (40 nM) AS-CDKL5 phosphorylates 200 ng (170 nM) EB2 very rapidly (E, F). In 30 minutes of incubation, 200 ng EB2 is phosphorylated by small amounts of CDKL5 (G, H)."

5. Was MAP1S S812 detected? Based on Fig 1K, S812 seems to be a more prevalent phosphorylation site than S786. Could you develop a targeted MS assay to quantify this site? *MAP1S S812 was not detected in the chemical genetic substrate screen. The reviewer rightly points out that all other evidence indicates this site is the preferred phosphorylation site by CDKL5. In silico trypsin digest (ExPASy PeptideCutter) shows that mouse MAP1S S812 is located in a small 5 amino acid peptide with MW 542 Da. The m/z of the double charged precursor ion is therefore outside of the m/z range typically used for MS1. We think a targeted MS assay would therefore likely be more effective using a different enzyme and could be a useful tool for more quantitative assays. For this manuscript, we did not attempt to perform further mass spectrometry analysis for this site. We feel that sufficient evidence was collected to establish MAP1S Ser812 as a genuine CDKL5 phosphorylation site: 1) MAP1S S812 phosphospecific antibody has been used to confirm this*

phosphorylation in vivo (Figure 2A-D) and 2) kinase assays using S812A phosphomutant MAP1S has been used to confirm phosphorylation of this site in vitro (Figure 1K).

6. Interesting correlation between increasing CDKL5 level and decreasing EB2 pS222 level as shown in Fig 2C. Can the authors provide additional investigation on how this is regulated?

We agree with the referee that this is an interesting correlation. Our results indicate a developmental regulation of EB2 pS222 that is independent of CDKL5 expression levels. Early postnatal development is a crucial time for neurons consisting of changes in various signalling pathways that may affect CDKL5 substrate phosphorylations. We continue to investigate this open question.

In this manuscript, we have provided evidence that EB2 phosphorylation is inhibited by NMDA receptor activity (Figure 3). Synaptogenesis in cortex takes place postnatally and is largely completed by P20. Increased NMDAR activation at synapses could potentially lead to the observed reduction of EB2 phosphorylation in later ages. This possibility is now stated in Discussion section on "CDKL5's regulation by neuronal activity": "It is possible that increased NMDA receptor activation upon maturation of synapses in adult mice inhibits baseline levels of EB2 pS222."

In this revised manuscript, we tested a second hypothesis with new experiments. We reasoned that increased phosphatase activity could result in reduced CDKL5 substrate phosphorylation later in development. "We treated cultured neurons at younger (DIV11) or older (DIV22) developmental stages with PP1/PP2A type phosphatase inhibitor, Okadaic acid (O.A.). We found that O.A. increases EB2 pS222 levels in both stages. The percent increase in EB2 pS222 was higher in DIV22 when compared to DIV11, indicating that PP1/PP2A activity may be higher in older stages than in younger neurons. However, even in the presence of O.A. the observed decrease in EB2 phosphorylation in older neurons remained, indicating that additional factors may contribute to observed reduction in pS222 (data now included in EV 3A & B)."

A corresponding discussion section was added "NMDAR signalling, phosphatase activity or developmental regulation of putative co-activators are among possible factors that reduce these phosphorylations in adult."

7. Critical data are missing from the evaluation of in-house generated EB2 pS222, MAP1S pS786, pS812 pAb in terms of the specificity and cross-reactivity to other proteins. This could raise question to Fig 2G, which seems like EB2 pS222 is present everywhere in cell (or abundant in nuclei), but EB2 is excluded from nuclei. This could either be: a) nuclei fraction EB2 is a small fraction in total protein, but highly phosphorylated, or b) the phospho antibody cross reacts with other proteins.

We thank the referee for this important comment. We now added blots for the full molecular weight ranges for EB2, EB2 pS222, MAP1S LC and MAP1S LC pS812 in WT and CDKL5 KO brain lysates (Figure EV2 D & E). In these blots, we point out the correct phospho-specific antibody bands as these bands' sizes match between total and phosphospecific protein blots and phosphorylation levels, detected by these phosphospecific antibodies, decrease in the KO brains. EB2 pS222 antibody recognizes two other non-specific bands and MAP1S pS812 recognizes numerous non-specific bands, however these do not interfere with the detection of the correct band, as exemplified here.

As referee rightly points out, we have presented EB2 pS222 in immunofluorescence stainings. In this context, it is critical to establish that detected signal belongs to EB2. In this version of our manuscript we added new experiments to address this issue. We used a previously reported shRNA sequence to knock-down endogenous EB2 in cultured neurons. A dramatic reduction in total EB2 as well as EB2 pS222 was observed in neuronal dendrites, in contrast to scrambled shRNA control (new Figure EV2 F). Interestingly, nuclear EB2 pS222 was not reduced upon EB2 shRNA expression. This finding and the lack of total EB2 staining in nucleus clearly indicate that nuclear staining observed with EB2 pS222 is non-specific background. This information is now included in Results section "MAP1S and EB2 are physiological substrates of CDKL5 in brain".

8. Fig 2C the MAP1S pS812 band is so low that hardly can make conclusion. Fig 2D is not clear whether it's quantifying CDKL5 WT or KO. Two genotypes but only one set of plot is shown. We agree with the reviewer that MAP1S pS812 is indeed a weak antibody and the full blots are now shown as stated above. We ensured to analyse the pS812 signal as accurately as possible and the

results are shown. We feel that despite signal being weak, the quantifications (consisting of multiple repeats) are accurate.

The quantification shown in Figure 2D is the CDKL5-dependent phosphorylation of EB2 and MAP1S and therefore consists of the WT signal after KO signal is subtracted from it. This has been clarified by changing the y-axis label to "CDKL5-dependent signal (%)" and adding an explanation to the legend of Figure 2C, D: "CDKL5-dependent substrate phosphorylation is quantified by subtracting KO from WT levels for each age."

9. Why MAP1S shRNA knock down has different conclusion in Fig 4F (n.s.) vs. in Fig EV 5E (**)? Why different statistics was used in Fig 4F (Tukey's multiple comparison) vs. in Fig EV 5E (Dunnet's multiple comparison)? More independent shRNA nucleotides may be needed to knock down the same target protein and show consistent result.

Different statistical analyses were used because of the difference in experimental set-up. In Figure 4F we compared the mean of each dataset with the mean of every other dataset (Tukey), while in EV Figure 5E the means of the shRNAs are compared to the control column (Dunnet). The different number of comparisons leads to different confidence intervals. For the purpose of this comment, we directly compared Control shRNA and MAP1S shRNA data in CDKL5 WT neurons reported on Figure 4F using a t-test. This test showed that these 2 data sets are significantly different. However, the total data sets analysed in Figure 4F consists of all 6 groups we report. To take into account all datasets we used Tukey, as reported in our paper. The added variance due to more data sets lead to no significance between Control and MAP1S shRNA data in this analysis.

In addition, in Figure 4F we analysed the data using the average EB3 comet lifetime per cell, while in our first version of EV Figure 5E we used the individual EB3 comet lifetimes. For consistency, we have now changed EV Figure 5E to depict the average EB3 comet lifetime per neuron, similar to Figure 4F. Both MAP1S shRNAs robustly knockdown overexpressed MAP1S in HEK293 cells and independently affect microtubule dynamics. We therefore do not believe that more shRNAs would provide additional value.

2nd Editorial Decision

31st Aug 2018

Thank you for submitting your revised manuscript for our consideration. It has now been seen once more by two of the original reviewers, whose comments are copied below. I am happy to inform you that all three are satisfied with the revisions and now unconditionally recommend publication in The EMBO Journal.

REFEREE REPORTS:

Referee #1:

The manuscript by Baltussen and colleagues has been modified extensively, and extra experiments have been performed. In response to my comments, the authors have:

- 1) Explained why TrkB-RFP was used as a model cargo, and added a statement to the manuscript
- 2) Demonstrated that S786/812Asp double mutant behaves like S786/812Ala, and although a double Glu mutant was not made, provided a logical reason why the Asp might fail to function as a phosphomimetic. They also confirm that MAP1S is indeed phosphorylated at appropriate sites, 786 and 812 in the fractions for the pull down, as requested
- 3) As also discussed by Reviewer 3, more interpretation of the MS data was required, including the fragmentation method applied and more spectrum annotation. This data has now been placed into EV1 and labelled 'best identification spectra'. Assuming Referee 3 concurs, this seems a reasonable outcome.

Referee #3:

The authors have substantially improved their data presentation and interpretation in the revised manuscript. This has cleared my previous concerns and I have no additional questions.

Corresponding Author Name: Sila Ulanir and Alysson Muotri

Manuscript Number: EMBOJ-2018-99763R